# Differentially Private Cross-Silo Recommendation from Implicit Feedback

Xun Ran[1]   Qingqing Ye[1]   Xin Huang[2]   Jianliang Xu[2]   Haibo Hu[1]

## Abstract

Cross-silo recommendation from implicit feedback is a key task in modern recommender systems, where user-item interaction data are distributed across multiple parties and cannot be centrally collected. Unlike explicit feedback, which provides fully observed real-valued ratings, implicit feedback is one-class and extremely sparse, recording only users' actions or inactions (e.g., clicks, visits, or bookmarks), yet it is far more prevalent in real-world applications. Such behavioral data are often highly sensitive, raising significant privacy concerns when used for collaborative model training. Although differential privacy (DP) has been widely applied to explicit feedback-based models, the resulting utility degradation makes it difficult to apply DP effectively to implicit feedback learning. In this work, we propose DPIMF, a differentially private implicit matrix factorization framework for cross-silo recommendation based on objective perturbation. To improve utility, we redesign the loss function and adopt an importance sampling scheme to reduce the noise scale required for privacy preservation. We further provide formal utility guarantees for the proposed techniques and characterize the conditions under which utility improvements are maximized. Experiments on three benchmark datasets validate our theoretical analysis and demonstrate that DPIMF achieves a better privacy-utility trade-off than state-of-the-art methods.

## 1. Introduction

Cross-silo collaborative recommendation is a fundamental problem with broad real-world applications (Kalloori

[1]Department of Electrical and Electronic Engineering, The Hong Kong Polytechnic University, Hong Kong SAR, China [2]Department of Computer Science, The Hong Kong Baptist University, Hong Kong SAR, China. Correspondence to: Qingqing Ye <qqing.ye@polyu.edu.hk>.

*Proceedings of the 43rd International Conference on Machine Learning*, Seoul, South Korea. PMLR 306, 2026. Copyright 2026 by the author(s).

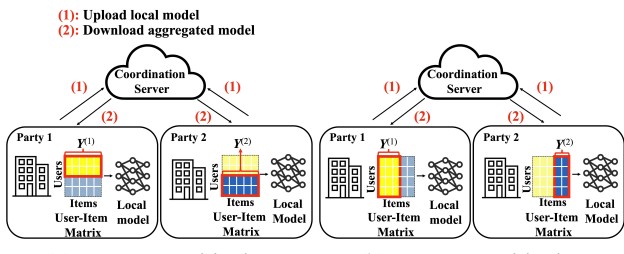

(a) User set partitioning   (b) Item set partitioning

*Figure 1.* Cross-silo recommendation settings.

& Klingler, 2021; Liu et al., 2022). In this setting, the user-item interaction matrix is distributed across multiple organizations (silos) that cannot directly share raw data. As illustrated in Figure 1, such partitioning typically follows two paradigms (Yang et al., 2020): (i) user set partitioning, where parties share a common item set but hold disjoint user subsets (e.g., regional branches of a platform), and (ii) item set partitioning, where parties share a common user set but hold disjoint item subsets (e.g., multiple applications on the same device). In both cases, the goal is to collaboratively learn from the global interaction matrix without exposing raw user data to other parties or the coordinating server (Li et al., 2021; Yang et al., 2019).

Matrix factorization (MF) naturally decomposes user and item representations, making it well aligned with cross-silo data partitioning while remaining scalable and effective (Li et al., 2021; Rendle et al., 2021; 2022; Koren et al., 2021). Classical recommender systems often rely on explicit feedback, represented as real-valued ratings (e.g., 1-5 stars). However, in many practical applications, feedback is implicit, recorded as binary interaction signals such as clicks, purchases, or page views. Implicit feedback is significantly more prevalent and easier to collect, and has therefore attracted increasing attention (Pan et al., 2008; Lian et al., 2018; Chen et al., 2023).

Although raw interaction data are never directly shared, model updates exchanged during collaborative training can still leak sensitive information (Fredrikson et al., 2015; Shokri et al., 2017). Differential privacy (DP) provides a principled framework to mitigate such leakage by bounding the information revealed through shared outputs. Existing DP-based recommender systems, however, primarily focus on explicit feedback-based MF (EMF) (Berlioz et al., 2015; Hua et al., 2015; Chien et al., 2021; Li et al., 2021). These

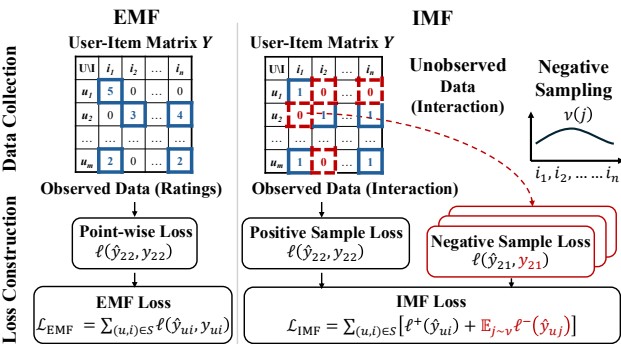

*Figure 2.* Comparison of EMF and IMF.

approaches do not directly extend to implicit feedback-based MF (IMF) due to fundamental differences in data structure and learning objectives.

Figure 2 highlights the key distinctions between EMF and IMF. In EMF, feedback is fully observed and real-valued, and the learning objective can be formulated as a point-wise regression loss $\mathcal{L}_{\text{EMF}}$, where $y_{ui} \in \mathbb{R}$ denotes the observed rating and $\ell$ is a point-wise loss function. Each training instance contributes independently to the loss, and modifying a single data point affects only its corresponding term.

In contrast, IMF operates in a one-class setting where only positive interactions are observed (Pan et al., 2008). Learning therefore relies on *negative sampling* to construct the loss $\mathcal{L}_{\text{IMF}}$, where negative items are sampled from a distribution $\nu(j)$. Unlike EMF, each data point in IMF may influence the loss both directly and indirectly through the sampling space, inducing complex interdependencies among training instances. From a differential privacy perspective, such interdependencies substantially complicate sensitivity analysis. Since sensitivity directly determines the noise magnitude required to achieve DP guarantees, IMF generally exhibits higher and more difficult-to-control sensitivity than EMF. This poses significant challenges for applying standard DP mechanisms.

Existing approaches attempt to address this issue by introducing DP at different stages of the learning process (Berlioz et al., 2015; Shin et al., 2018; Gao et al., 2020; Guo et al., 2019; Li et al., 2021). Input perturbation methods inject noise directly into the interaction data (Gao et al., 2020; Guo et al., 2019), but implicit feedback is extremely sparse and high-dimensional, making such approaches prone to severe utility degradation. Gradient perturbation methods add calibrated noise to gradients during training (Shin et al., 2018; Li et al., 2021; Abadi et al., 2016), but in IMF, the dependence on negative sampling leads to noise accumulation across iterations, resulting in slow convergence and degraded model quality.

In this work, we study differentially private implicit matrix factorization (DPIMF) in the cross-silo setting. We adopt an objective perturbation framework, which injects noise once into the loss function, thereby avoiding the excessive noise accumulation inherent to input and gradient perturbation methods. Designing such a method, however, is nontrivial. The privacy mechanism must protect a user's interaction profile, including both observed and unobserved behaviors, while the extreme sparsity of implicit data can lead to prohibitively large noise scales. Excessive noise may further distort the loss landscape, resulting in unstable optimization and poor recommendation utility.

To address these challenges, we propose three strategies. First, we redesign the negative sampling and loss formulation of IMF to reduce DP sensitivity by leveraging the offsetting structure between observed and unobserved terms. Second, we introduce spectral regularization to keep the noisy loss well-behaved by restoring the positive definiteness of its (already symmetric) coefficient matrix. Third, we incorporate importance sampling to achieve privacy amplification, which reduces the required noise without additional privacy loss while preserving informative training signals. We theoretically show that the proposed sampling scheme yields effective privacy amplification. Our contributions are summarized as follows:

- We present DPIMF, the first framework for differentially private implicit matrix factorization in cross-silo recommendation. By redesigning the loss function to suppress sensitivity and applying spectral regularization to keep the privatized objective bounded, DPIMF preserves model utility.
- We integrate importance sampling into DP-enabled MF and theoretically establish its privacy amplification effect, achieving simultaneous improvements in privacy and accuracy.
- We provide utility guarantees and rigorous privacy analysis, proving that the proposed schemes satisfy pure $\varepsilon$-DP.
- Extensive experiments on real-world datasets demonstrate that DPIMF consistently outperforms state-of-the-art baselines in recommendation accuracy.

The remainder of this paper is organized as follows. In Section 2, we introduce preliminaries and the studied problem. Sections 3 and 4 present the proposed schemes in detail and Section 5 provides the theoretical analysis of the proposed schemes. The experimental results are presented and analyzed in Section 6. The related work is discussed in Section 7. Section 8 concludes the paper.

## 2. Preliminaries and Problem Formulation

### 2.1. Implicit Matrix Factorization

Implicit matrix factorization models user preferences from binary implicit feedback, where each entry records whether a user has interacted with an item. The feedback matrix

is highly sparse, with most entries unobserved. Because an unobserved entry does not necessarily indicate negative preference, learning typically relies on jointly modeling observed interactions and a sampled subset of unobserved ones.

Let $U$ and $I$ denote the sets of users and items, respectively. The implicit feedback is represented by a binary matrix $Y \in \{0,1\}^{|U| \times |I|}$, where $y_{ui} = 1$ indicates that user $u \in U$ interacted with item $i \in I$, and $y_{ui} = 0$ otherwise. IMF learns latent representations $P \in \mathbb{R}^{d \times |U|}$ and $Q \in \mathbb{R}^{d \times |I|}$, where $d$ is the latent dimension. Specifically, $p_u \in \mathbb{R}^d$ and $q_i \in \mathbb{R}^d$ denote the latent embeddings of user $u$ and item $i$, corresponding to the $u$-th and $i$-th columns of $P$ and $Q$, respectively. The predicted interaction score is given by $\hat{y}_{ui} = p_u^\top q_i$. A general IMF objective can be written as

$$L(P, Q) = \sum_{(u,i) \in \mathcal{S}} [\ell^+(\hat{y}_{ui}) + \mathbb{E}_{j \sim \nu(u)} \, \ell^-(\hat{y}_{uj})], \quad (1)$$

where $\mathcal{S} \subseteq U \times I$ denotes the set of observed interactions. Here, $\ell^+(\cdot)$ and $\ell^-(\cdot)$ are the loss functions for observed and unobserved interactions, respectively, and $\nu(u)$ is a user-specific sampling distribution over items (Pan et al., 2008).

In practice, the expectation over unobserved interactions is approximated via *negative sampling*, whose choice directly affects learning efficiency and, in private settings, the required noise scale. A regularization term $R(P, Q)$ is added to control model complexity, and the resulting objective can be optimized using standard methods such as stochastic gradient descent or alternating optimization (Berlioz et al., 2015).

## 2.2. Differential Privacy

Differential privacy ensures that the output of an algorithm is insensitive to the presence or absence of any single data record, even in the presence of auxiliary information (Dwork et al., 2006). In this work, we adopt a *record-level* privacy notion, where two datasets are neighboring if they differ in a single user-item interaction, and we target pure $\varepsilon$-DP—the strictest DP guarantee. This rigor is afforded by our one-shot perturbation mechanism: unlike iterative methods that resort to the relaxed $(\varepsilon, \delta)$-DP for a tighter composition accountant, we perturb each objective only once and thus incur no such composition cost.

*Definition* 1 ($\varepsilon$-Differential Privacy). An algorithm $\mathcal{A}$ satisfies $\varepsilon$-DP if for any neighboring datasets $D$ and $D'$ and any measurable output set $O$,

$$\Pr[\mathcal{A}(D) \in O] \leq \exp(\varepsilon) \Pr[\mathcal{A}(D') \in O]. \quad (2)$$

**Sensitivity and Laplace mechanism.** For a function $f$, its $\ell_1$-sensitivity is defined as

$$\Delta_f = \max_{D \sim D'} \|f(D) - f(D')\|_1, \quad (3)$$

---

**Algorithm 1** A Paradigm for DPIMF (Solving Private Item Profile Matrix)

**Input:** $\{S_1, S_2, \ldots, S_K\}$ owned by $K$ parties; total iterations $T$.
**Output:** $P$; $\bar{Q}$.

*// Phase 1: Initialization*
1: Server initializes item profile matrix $Q$.
2: Each party $k$ initializes local user profiles $P_{U^{(k)}}$.
  *// Phase 2: Collaborative Learning*
3: **for** $t = 1$ **to** $T$ **do**
    *// Local Update (in each party)*
4:  For each $i \in I^{(k)}$ in parallel, form the per-item objective $L_k(q_i)$.
5:  Apply an objective perturbation mechanism to obtain a privatized objective $\bar{L}_k(q_i)$ (details in Section 3).
6:  Compute $\bar{q}_i \leftarrow \arg\min_{\|q_i\|_2 \leq \sqrt{1/\lambda}} \bar{L}_k(q_i)$.
    *// Global Update*
7:  Server aggregates $\{\bar{Q}_{I^{(1)}}, \ldots, \bar{Q}_{I^{(K)}}\}$ and broadcasts the result.
8: **end for**
  *// Phase 3: Local Fine-tuning*
9: Each party refines $P_{U^{(k)}}$ using the aggregated item profiles.

---

where $D \sim D'$ denotes neighboring datasets. The Laplace mechanism releases

$$\mathcal{A}(D) = f(D) + \eta, \quad (4)$$

where each coordinate of $\eta$ is drawn i.i.d. from $\mathrm{Lap}(\Delta_f/\varepsilon)$.

## 2.3. Problem Formulation

We consider cross-silo recommendation with $K$ parties and an honest-but-curious coordination server (Li et al., 2021; Hua et al., 2015; Chien et al., 2021). The interaction data are distributed across parties, either by users (user set partitioning) or by items (item set partitioning), and no party shares raw interaction records.

Let $D = \{S_1, \ldots, S_K\}$ denote the distributed dataset, where $S_k$ is the set of observed interactions held by party $k$. Let $U^{(k)} = \{u \mid (u,i) \in S_k\}$ and $I^{(k)} = \{i \mid (u,i) \in S_k\}$. Each party constructs a local objective $L_k(P_{U^{(k)}}, Q_{I^{(k)}})$ based on Eq. (1). The global objective aggregates local objectives as

$$L(P, Q) = \sum_{k=1}^{K} \pi_k \, L_k(P_{U^{(k)}}, Q_{I^{(k)}}), \quad (5)$$

where $\pi_k$ is the weight of party $k$.

Under this honest-but-curious model, both the parties and the server follow the protocol but may attempt to infer sensitive information from the exchanged messages. Our goal is to learn and communicate model parameters under rigorous $\varepsilon$-DP while preserving recommendation utility.

## 3. A Strawman Solution for DPIMF

This section introduces a general paradigm for solving DPIMF in the cross-silo setting and presents a strawman implementation based on objective perturbation. Since the

user and item profile matrices appear symmetrically in the loss function of Eq. (1), they can be learned by solving the same optimization problem. Without loss of generality, we focus on learning the private item profiles $\bar{Q}$; the user profiles $\bar{P}$ can be obtained analogously.

As summarized in Algorithm 1, the DPIMF paradigm consists of three phases: (i) initialization of global item profiles at the server and local user profiles at each party (Lines 1-2); (ii) collaborative learning, where each party performs client-side private updates via objective perturbation and the server aggregates the privatized item profiles through post-processing (Lines 3-7); and (iii) local fine-tuning of user profiles using the aggregated item profiles (Line 9). We next detail the client-side local update procedure.

**Local objective via uniform negative sampling.** For an item $i \in I$, let $U_i \subseteq U$ denote the users who interacted with $i$, where $U$ is the entire user set. We adopt *uniform negative sampling* over $U$ to approximate the contribution of unobserved interactions, which yields an unbiased estimator of the full implicit loss while ensuring that every user contributes equally to the negative signal (Rendle et al., 2021; 2022). Taking expectation over the sampled negatives leads to the following per-item objective:

$$L^O(\boldsymbol{q}_i) = \sum_{u \in \boldsymbol{U}_i} \left(\boldsymbol{p}_u^\top \boldsymbol{q}_i - 1\right)^2 + \alpha_0 \sum_{u \in \boldsymbol{U}} \left(\boldsymbol{p}_u^\top \boldsymbol{q}_i\right)^2$$
$$+ \lambda\left(|\boldsymbol{U}_i| + \alpha_0|\boldsymbol{U}|\right)\|\boldsymbol{q}_i\|_2^2, \quad (6)$$

where the last term is a frequency-based regularizer that penalizes frequently interacted items more heavily to prevent overfitting.

**Coefficient expansion.** Direct sensitivity analysis of Eq. (6) with respect to changes in individual user interactions is challenging due to its coupled summation structure. To facilitate analysis, we rewrite the objective as a quadratic form by expanding all terms and discarding constants independent of $\boldsymbol{q}_i$. Specifically, Eq. (6) can be expressed as

$$l(\boldsymbol{q}_i) = \boldsymbol{q}_i^\top \left(\boldsymbol{G}_{\boldsymbol{U}_i} + \alpha_0 \boldsymbol{G}_{\boldsymbol{U}}\right)\boldsymbol{q}_i - 2\boldsymbol{q}_i^\top \boldsymbol{g}_{\boldsymbol{U}_i} + \rho_i\|\boldsymbol{q}_i\|_2^2, \ (7)$$

where, for any user subset $\mathcal{V} \subseteq \boldsymbol{U}$, $\boldsymbol{G}_{\mathcal{V}} = \sum_{u \in \mathcal{V}}(\boldsymbol{p}_u \otimes \boldsymbol{p}_u)$, $\boldsymbol{g}_{\mathcal{V}} = \sum_{u \in \mathcal{V}} \boldsymbol{p}_u$, and $\rho_i = \lambda(|\boldsymbol{U}_i| + \alpha_0|\boldsymbol{U}|)$ is the regularization coefficient inherited from Eq. (6). Here $\otimes$ denotes the outer product: for $\boldsymbol{x}, \boldsymbol{y} \in \mathbb{R}^d$, $\boldsymbol{x} \otimes \boldsymbol{y} \in \mathbb{R}^{d \times d}$ is the rank-one matrix with entries $(\boldsymbol{x} \otimes \boldsymbol{y})_{ab} = x_a y_b$.

**Objective perturbation and sensitivity.** Changing a single implicit interaction associated with item $i$ affects only the corresponding user set $\boldsymbol{U}_i$. Let $\boldsymbol{U}_i' = \boldsymbol{U}_i \cup \{v\}$ denote a neighboring dataset, where for a matrix $\boldsymbol{A} \in \mathbb{R}^{d \times d}$ we use the entry-wise $\ell_1$ norm $\|\boldsymbol{A}\|_{1,1} \triangleq \sum_{a=1}^d \sum_{b=1}^d |A_{ab}|$. Bounding the $\ell_1$ difference of the polynomial coefficients in Eq. (7) then yields the sensitivity

$$\Delta = \max_{u \in \boldsymbol{U}} \left(2\|\boldsymbol{p}_u\|_1 + \|\boldsymbol{p}_u \otimes \boldsymbol{p}_u\|_{1,1} + 1\right). \quad (8)$$

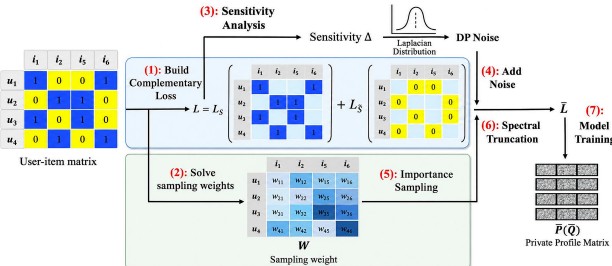

*Figure 3.* Overview of the full-fledged DPIMF.

To guarantee $\varepsilon$-differential privacy, we perturb the coefficients of $l(\boldsymbol{q}_i)$ using Laplace noise. The resulting private objective is

$$\bar{l}(\boldsymbol{q}_i) = \boldsymbol{q}_i^\top \left(\boldsymbol{G}_{\boldsymbol{U}_i} + \alpha_0 \boldsymbol{G}_{\boldsymbol{U}} + \boldsymbol{B}\right)\boldsymbol{q}_i$$
$$- \boldsymbol{q}_i^\top \left(2\boldsymbol{g}_{\boldsymbol{U}_i} + \boldsymbol{b}\right) + (\rho_i + \lambda\eta)\|\boldsymbol{q}_i\|_2^2, \quad (9)$$

where $\boldsymbol{B} \in \mathbb{R}^{d \times d}$, $\boldsymbol{b} \in \mathbb{R}^d$, and $\eta \in \mathbb{R}$ are independently drawn from $\text{Lap}(\Delta/\varepsilon)$.

**Private optimization.** The non-private minimizer satisfies $\|\boldsymbol{q}_i^*\|_2 \leq \sqrt{1/\lambda}$ (Lemma 1, proved in App. E). We therefore compute the private profile by solving $\bar{\boldsymbol{q}}_i = \arg\min_{\|\boldsymbol{q}_i\|_2 \leq \sqrt{1/\lambda}} \bar{l}(\boldsymbol{q}_i)$. Repeating this procedure for all items yields the private item profile matrix $\bar{Q}$.

**Remark.** Let $c = \max_{u,j} |p_{uj}|$. Since $\|\boldsymbol{p}_u\|_1 \leq cd$ and $\|\boldsymbol{p}_u \otimes \boldsymbol{p}_u\|_{1,1} \leq c^2 d^2$, the sensitivity in Eq. (8) scales as $O(c^2 d^2)$. This large noise scale leads to substantial utility degradation, motivating the refined design in the next section.

## 4. The Full-fledged DPIMF

The strawman solution in Section 3 yields a valid differentially private pipeline but incurs high sensitivity due to uniformly treating all unobserved interactions. We propose a refined design that substantially reduces sensitivity while preserving model fidelity by explicitly linking negative sampling choices to loss coefficients and sensitivity.

Our approach reduces the required noise scale through three key ideas: (i) redesigning the loss to better balance sensitivity and utility, (ii) regularizing quadratic coefficients to ensure boundedness of the privatized objective, and (iii) introducing a sampling strategy that amplifies privacy while retaining informative signals. Accordingly, we extend the strawman solution with three components: *complementary loss construction*, *spectral truncation*, and *importance sampling*. As illustrated in Figure 3, we construct the complementary loss, analyze its sensitivity, compute importance weights, and optimize a noise-perturbed, spectrally truncated objective. Algorithmic details and the extension to user profiles are deferred to App. A.

### 4.1. Complementary Sampling for Sensitivity Control

Negative sampling plays a central role in implicit matrix factorization. In Section 3, unobserved interactions are uniformly treated as negatives, yielding an objective term that aggregates over all users and induces large-magnitude coefficients. We instead adopt *complementary negative sampling*, which assigns a tunable weight to a complementary negative set. This construction induces coefficient-level cancellation and directly reduces sensitivity.

**Complementary loss construction.** For item $i$, let $\boldsymbol{U}_i$ be the users with observed interactions and $\tilde{\boldsymbol{U}}_i = \boldsymbol{U} \setminus \boldsymbol{U}_i$ the complementary unobserved set. We define the per-item loss as

$$L(\boldsymbol{q}_i) = \sum_{u \in \boldsymbol{U}_i} (\boldsymbol{p}_u^\top \boldsymbol{q}_i - 1)^2 + \alpha_0 \sum_{u \in \tilde{\boldsymbol{U}}_i} (\boldsymbol{p}_u^\top \boldsymbol{q}_i)^2 \qquad (10)$$
$$+ \lambda \big(|\boldsymbol{U}_i| + \alpha_0 |\tilde{\boldsymbol{U}}_i|\big) \|\boldsymbol{q}_i\|_2^2,$$

which differs from Eq. (6) in that the negative term is restricted to $\tilde{\boldsymbol{U}}_i$ and scaled by $\alpha_0$.

This modification preserves predictive accuracy under mild assumptions (Theorem 3) while fundamentally altering the sensitivity of the objective.

**Sensitivity reduction.** Expanding Eq. (10) and omitting constant terms yields

$$l(\boldsymbol{q}_i) = \boldsymbol{q}_i^\top \big(\boldsymbol{G}_{\boldsymbol{U}_i} + \alpha_0 \boldsymbol{G}_{\tilde{\boldsymbol{U}}_i}\big) \boldsymbol{q}_i$$
$$- 2\boldsymbol{q}_i^\top \boldsymbol{g}_{\boldsymbol{U}_i} + \lambda \big(|\boldsymbol{U}_i| + \alpha_0 |\tilde{\boldsymbol{U}}_i|\big) \|\boldsymbol{q}_i\|_2^2. \qquad (11)$$

Using the identity $\boldsymbol{G}_{\tilde{\boldsymbol{U}}_i} = \boldsymbol{G}_{\boldsymbol{U}} - \boldsymbol{G}_{\boldsymbol{U}_i}$, the quadratic coefficient can be rewritten as

$$\boldsymbol{G}_{\boldsymbol{U}_i} + \alpha_0 \boldsymbol{G}_{\tilde{\boldsymbol{U}}_i} = \alpha_0 \boldsymbol{G}_{\boldsymbol{U}} + (1 - \alpha_0) \boldsymbol{G}_{\boldsymbol{U}_i}. \qquad (12)$$

This decomposition yields the first source of sensitivity reduction: only the item-dependent term $\boldsymbol{G}_{\boldsymbol{U}_i}$ is affected by a single interaction, and its contribution is scaled by $(1 - \alpha_0)$, while $\alpha_0 \boldsymbol{G}_{\boldsymbol{U}}$ is item-independent. A second reduction arises from symmetry. Since each $\boldsymbol{G}_{\mathcal{V}} = \sum_{u \in \mathcal{V}} \boldsymbol{p}_u \otimes \boldsymbol{p}_u$ is symmetric, it suffices to perturb only the upper triangular entries of the quadratic coefficient, enforcing symmetry by mirroring. For neighboring datasets $\boldsymbol{U}_i' = \boldsymbol{U}_i \cup \{v\}$, the resulting sensitivity of the quadratic term satisfies

$$\Delta_2 \leq \frac{c^2 d (d+1)(1 - \alpha_0)}{2}, \qquad (13)$$

where the factor $1/2$ follows from triangular perturbation. Similarly, the regularization coefficient $|\boldsymbol{U}_i| + \alpha_0 |\tilde{\boldsymbol{U}}_i|$ changes by at most $(1 - \alpha_0)$, yielding $\Delta_3 \leq (1 - \alpha_0)$. The linear term is unaffected, with $\Delta_1 = \max_{u \in \boldsymbol{U}} 2\|\boldsymbol{p}_u\|_1 \leq 2cd$. Overall, sensitivity reduction stems from (i) coefficient cancellation via complementary negative sampling, and (ii) symmetric structure of the quadratic term. The detailed sensitivity analysis is deferred to App. L.

### 4.2. Private Objective and Learning

Based on the sensitivities derived above, we split the privacy budget $\varepsilon$ into three parts, $\varepsilon_k = \varepsilon \beta_k$ with $\sum_{k=1}^3 \beta_k = 1$, corresponding to the linear, quadratic, and regularization terms. We inject Laplace noise calibrated to each sensitivity and construct the private objective:

$$\bar{l}(\boldsymbol{q}_i) = \boldsymbol{q}_i^\top \boldsymbol{M}_i \boldsymbol{q}_i - \boldsymbol{q}_i^\top \boldsymbol{h}_i + \lambda_i \|\boldsymbol{q}_i\|_2^2, \qquad (14)$$

where $\boldsymbol{M}_i = \alpha_0 \boldsymbol{G}_{\boldsymbol{U}} + (1 - \alpha_0) \boldsymbol{G}_{\boldsymbol{U}_i} + \tilde{\boldsymbol{B}}$, $\boldsymbol{h}_i = 2\boldsymbol{g}_{\boldsymbol{U}_i} + \boldsymbol{b}$, and $\lambda_i = \lambda(|\boldsymbol{U}_i| + \alpha_0 |\tilde{\boldsymbol{U}}_i| + \eta)$. Here $\boldsymbol{b} \in \mathbb{R}^d$, $\tilde{\boldsymbol{B}} \in \mathbb{R}^{d \times d}$, and $\eta \in \mathbb{R}$ are drawn independently from $\mathrm{Lap}(\Delta_1 / \varepsilon_1)$, $\mathrm{Lap}(\Delta_2 / \varepsilon_2)$, and $\mathrm{Lap}(\Delta_3 / \varepsilon_3)$, respectively, with $\tilde{\boldsymbol{B}}$ symmetrized by mirroring its upper triangle. The private item profile $\bar{\boldsymbol{q}}_i$ is obtained by minimizing $\bar{l}(\boldsymbol{q}_i)$ over the constraint set $\{\boldsymbol{q}_i : \|\boldsymbol{q}_i\|_2 \leq \sqrt{1/\lambda}\}$. Repeating this for all items yields the private item profile matrix $\bar{\boldsymbol{Q}}$.

### 4.3. Spectral Truncation

Although complementary loss reduces noise, directly perturbing the loss coefficients may render the objective unbounded. After noise injection, the privatized objective takes the quadratic form $\bar{l}(\boldsymbol{q}_i) = \boldsymbol{q}_i^\top \boldsymbol{M} \boldsymbol{q}_i - \boldsymbol{q}_i^\top \boldsymbol{h} + \tau$, where $\boldsymbol{M} = \boldsymbol{M}_i + \lambda_i \boldsymbol{E}$ folds the regularization into the quadratic coefficient ($\boldsymbol{E}$ is the $d \times d$ identity), $\boldsymbol{h} = \boldsymbol{h}_i$, and $\tau$ collects the constant terms; this form is bounded if $\boldsymbol{M}$ is symmetric positive definite.

Symmetry is ensured by construction, but positive definiteness may be violated due to noise. To address this, we perform eigendecomposition $\boldsymbol{M} = \boldsymbol{V} \Lambda \boldsymbol{V}^\top$ and replace each non-positive eigenvalue $\Lambda_{kk}$ by a small constant $\xi$: $\Lambda_{kk}' = \max(\Lambda_{kk}, \xi)$. Reconstructing $\boldsymbol{M}' = \boldsymbol{V} \Lambda' \boldsymbol{V}^\top$ yields a bounded and convex objective.

This *spectral truncation* step preserves $\varepsilon$-DP by postprocessing and has minimal impact on utility, as only noise-induced components are modified.

### 4.4. Importance Sampling

After the complementary loss, only the observed interactions remain privacy-sensitive, since the unobserved ones are absorbed into the zero-sensitivity aggregate $\boldsymbol{G}_{\boldsymbol{U}}$. We thus amplify their privacy with a weighted Poisson sampler, termed *Poisson importance sampling*, that retains each observed $(u, i)$ with probability $1/w_{ui}$ and reweights it by $w_{ui}$. As $\alpha_0 \boldsymbol{G}_{\boldsymbol{U}}$ is never subsampled, the offsetting (Eq. (12)) holds for every draw. This induces a privacy loss profile $\phi$.

*Theorem* 1 (Privacy Amplification). For any $\varepsilon$-DP mechanism $\mathcal{M}$ and weights $\{w_{ui}\}_{(u,i) \in \boldsymbol{U} \times \boldsymbol{I}}$, composing $\mathcal{M}$ with the Poisson importance sampler $\mathcal{S}_W$ yields a mechanism that satisfies $\varepsilon'$-DP with

$$\varepsilon' = \max_{(u,i) \in \boldsymbol{U} \times \boldsymbol{I}} \ln \bigg(1 + \frac{1}{w_{ui}} \big(e^{\phi(w_{ui}, u, i)} - 1\big)\bigg),$$

i.e., the worst-case per-pair privacy loss attained by that weight assignment.

*Proof:* See App. F. ∎

Conversely, to meet a *prescribed* budget $\varepsilon'$ at minimum expected sample size, we invert this guarantee and optimize the weights via a convex program (Problem 1); the per-pair constraint binds at the optimum, so the achieved budget equals the target. Crucially, the optimal weights $\boldsymbol{W}^*$ depend only on $\phi$ and the target budget $\varepsilon'$, not on the private interactions, so the sampler is data-independent and adds no privacy side-channel beyond the amplification above. Details and proofs are provided in App. B and App. C.

## 5. Theoretical Analysis

In this section, we establish the privacy and utility guarantees of the proposed DPIMF framework.

### 5.1. Privacy Analysis

*Theorem 2.* DPIMF satisfies $\varepsilon$-differential privacy.

*Proof:* We provide a proof sketch here and defer the complete proof to App. G. Based on the sensitivity analysis in Section 4, solving the private optimization problem $\bar{\boldsymbol{q}}_i = \arg\min \bar{l}(\boldsymbol{q}_i)$ in Eq. (14) for any fixed item $i$ satisfies $\varepsilon$-DP by the Laplace mechanism.

Moreover, in the cross-silo setting, the local objectives corresponding to different items and users are defined over disjoint subsets of interaction records. By the parallel composition property of differential privacy (Dwork et al., 2014), the joint release of the entire private profile matrices $\bar{\boldsymbol{Q}}$ and $\bar{\boldsymbol{P}}$ also satisfies $\varepsilon$-DP. ∎

### 5.2. Utility Analysis

We now analyze the utility of the proposed DPIMF methods. To control sensitivity and improve utility, we replace the original loss in Eq. (6) with the complementary loss in Eq. (10). The following theorem shows that this modification introduces only a bounded deviation in the optimal objective value.

*Theorem 3.* Let $N = \sqrt{1/\lambda}$ and $c = \max_{u \in \boldsymbol{U}, \, j \in [d]} |p_{uj}|$. Let $\boldsymbol{q}_i^* = \arg\min L(\boldsymbol{q}_i)$ and $\boldsymbol{q}_i' = \arg\min L^{\mathrm{O}}(\boldsymbol{q}_i)$ denote the minimizers of the redesigned loss in Eq. (10) and the original loss in Eq. (6), respectively. Then,

$$L(\boldsymbol{q}_i^*) - L^{\mathrm{O}}(\boldsymbol{q}_i') \leq \alpha_0 |\boldsymbol{U}_i| N^2 (\lambda + 2dc^2). \quad (15)$$

*Proof:* See App. H. ∎

Theorem 3 shows that the minimizer of the redesigned loss remains close to that of the original loss, with the gap controlled by $\alpha_0$, $\lambda$, and the latent dimension $d$. Empirical results in Section 6 further confirm that this deviation is negligible compared to the noise introduced by differential privacy.

**Utility comparison of different strategies.** To evaluate the effect of each design component in Section 4, we consider four strategies: $\mathcal{M}_1$, $\mathcal{M}_2$, $\mathcal{M}_3$, and $\mathcal{M}_4$. Specifically, $\mathcal{M}_1$ corresponds to the strawman objective perturbation in Eq. (9); $\mathcal{M}_2$ incorporates complementary loss with the full quadratic sensitivity $\Delta_2 = (1 - \alpha_0)c^2 d^2$; $\mathcal{M}_3$ further applies symmetric noise to the quadratic term, reducing it to the $\Delta_2$ in Eq. (13); and $\mathcal{M}_4$ sets $\alpha_0 = 1$ (making spectral truncation vacuous) and applies importance sampling.

*Theorem 4.* Let $N = \sqrt{1/\lambda}$, $\varepsilon_1 = \beta_1 \varepsilon$, $\varepsilon_2 = \beta_2 \varepsilon$ with $\beta_1 + \beta_2 \leq 1$, $c = \max_{u,j} |p_{uj}|$, and let $\mathcal{S}_W$ denote the importance sampler obtained from Eq. (17). Given the non-private loss $L(\boldsymbol{q}_i)$ in Eq. (10) and its minimizer $\boldsymbol{q}_i^*$, let $\bar{\boldsymbol{q}}_i^{(k)}$ denote the private minimizer produced by strategy $\mathcal{M}_k$, $k = 1, \ldots, 4$. Then the expected excess risk satisfies

$$\mathbb{E}\Big[L(\bar{\boldsymbol{q}}_i^{(1)}) - L(\boldsymbol{q}_i^*)\Big] \leq \frac{\sqrt{2}cdN\Big[(N + \frac{2\sqrt{d}}{d})(d+1)^2\Big]}{\varepsilon},$$

$$\mathbb{E}\Big[L(\bar{\boldsymbol{q}}_i^{(2)}) - L(\boldsymbol{q}_i^*)\Big] \leq \frac{\sqrt{2}cdN\Big[\frac{Nd^2(1-\alpha_0)}{\beta_2} + \frac{4\sqrt{d}}{\beta_1}\Big]}{\varepsilon},$$

$$\mathbb{E}\Big[L(\bar{\boldsymbol{q}}_i^{(3)}) - L(\boldsymbol{q}_i^*)\Big] \leq \frac{\sqrt{2}cdN\Big[\frac{Nd(1+d)(1-\alpha_0)}{2\beta_2} + \frac{4\sqrt{d}}{\beta_1}\Big]}{\varepsilon},$$

$$\mathbb{E}\Big[L(\bar{\boldsymbol{q}}_i^{(4)}) - L(\boldsymbol{q}_i^*)\Big] \leq \frac{4\sqrt{2d}\,cdN}{\varepsilon}. \quad (16)$$

*Proof:* See App. I. ∎

The above theorem bounds the expected excess risk of the four strategies for the proposed differentially private IMF. A higher bound indicates a larger loss in the utility of the strategy. Let $\gamma_1, \gamma_2, \gamma_3, \gamma_4$ denote the four upper bounds in Eq. (16). Their relationship is summarized below.

*Corollary 1.* Given $\beta_1, \beta_2 \in [0, 1]$, we have $\gamma_2 \geq \gamma_3 \geq \gamma_4$ and $\gamma_1 > \gamma_4$ for all $\alpha_0 \in [0, 1]$ and $d \in \mathbb{N}^+$. Moreover, $\gamma_1 > \gamma_2$ whenever $\alpha_0 \geq 1 - \beta_2$ and $d \geq \sqrt{1/\beta_1} - 1$.

*Proof:* See App. J. ∎

Corollary 1 implies that, with an appropriate $\alpha_0$, the complementary loss ($\mathcal{M}_2$) tightens the excess-risk bound over the strawman ($\mathcal{M}_1$); a larger $\alpha_0$ is needed when little budget is given to the quadratic term (small $\beta_2$), since $(1 - \alpha_0)$ scales its sensitivity. The gain from $\mathcal{M}_2$ to $\mathcal{M}_3$ comes from symmetrizing the perturbed quadratic coefficient, which reduces the effective noise dimension. The tightest bound is attained by $\mathcal{M}_4$ at $\alpha_0 = 1$, where the quadratic and regularization sensitivities vanish and require no perturbation; importance sampling then amplifies privacy while keeping the loss estimate unbiased, so the gains persist after sampling. These findings are validated in Section 6.

## 5.3. Time Complexity Analysis

We summarize the per-round cost of the full-fledged DPIMF in Section 4 (Algorithm 2) for updating the item profile matrix; the user side is symmetric. Let $n = |\boldsymbol{U}|$, $m = |\boldsymbol{I}|$, and $|\mathcal{S}| = \sum_i |\boldsymbol{U}_i|$ denote the numbers of users, items, and observed interactions, and let $d$ be the latent dimension. Combining the one-time preprocessing, the dominant per-item update cost $O(|\boldsymbol{U}_i| d^2 + d^3)$, and the importance-weight computation via Algorithm 3, one full round of alternating updates over items and users costs

$$O\big( |\mathcal{S}| d^2 + (n + m) d^3 + n \log(1/\rho) \big),$$

where $\rho$ is the bisection tolerance of the weight optimizer and the weights are per-user (one bisection per user). The full step-by-step derivation is deferred to App. D.

**Comparison with non-private IMF.** Standard implicit alternating least squares (Rendle et al., 2022) incurs $O(|\mathcal{S}| d^2 + (n + m) d^3)$ per round, where the $d^3$ term comes from solving a $d \times d$ linear system per user/item. DPIMF replaces each linear solve with an eigendecomposition of the same order and adds only the lightweight weight-optimization step $O(n \log(1/\rho))$. Hence the privacy mechanism preserves the asymptotic per-round complexity of the underlying optimizer, introducing only a constant-factor overhead from noise sampling and spectral truncation. Moreover, at the optimal strategy $\mathcal{M}_4$ ($\alpha_0 = 1$) we have $\Delta_2 = \Delta_3 = 0$, so the quadratic coefficient $\boldsymbol{M}$ stays positive semidefinite, the spectral-truncation step is vacuous, and the per-item cost reduces to the same $O(d^3)$ solve as non-private IMF.

## 6. Experimental Evaluation

In this section, we first introduce our experiment setup in Section 6.1, and then present the experimental results and our analysis in Section 6.2.

### 6.1. Experiment Setup

#### 6.1.1. DATASETS.

In the experiment, we evaluate our methods on three real-world datasets: **MovieLens 10M (ML-10M)** [1], **YahooMusic** [2], and **Amazon** [3]. ML-10M contains user-movie ratings from the GroupLens project. YahooMusic consists of user interactions with Yahoo's music service. Amazon captures user behavior on Amazon Instant Video. Dataset statistics are summarized in Table 1.

Following Section 2.3, we simulate local datasets by partitioning each dataset either by users (for item profile exchange) or by items (for user profile exchange), corresponding to the two cross-silo scenarios.

*Table 1.* The datasets used in our experiments.

| Property | ML-10M | YahooMusic | Amazon |
|---|---|---|---|
| Users | 71567 | 8089 | 5130 |
| Items | 10681 | 1000 | 1685 |
| Density | 4.3% | 3.4% | 0.7% |
| Avg. #interactions per user | 97 | 33 | 12 |
| Avg. #interactions per item | 40.5 | 270 | 36 |

#### 6.1.2. EVALUATION METRICS

We adopt the widely used leave-one-out evaluation, holding out each user's most recent interaction for testing and training models on the remaining data. Recommendation performance is measured by Hit Ratio (HR) and Normalized Discounted Cumulative Gain (NDCG), which evaluate whether the test item is recommended and its rank position, respectively. Higher HR and NDCG indicate better performance. Formal definitions are provided in App. K.3.

#### 6.1.3. COMPETITORS

We compare against the non-private ground truth and several state-of-the-art DP collaborative filters for implicit data: **GT** (Rendle et al., 2022), a non-private iALS baseline for Eq. (1); **DPMF** (Hua et al., 2015), objective-perturbation DP MF for explicit feedback; **DPLCF** (Gao et al., 2020) and **LDPICF** (Guo et al., 2019), nearest-neighbor methods that perturb feedback by random flipping and publish a sanitized item-similarity matrix; **SGDMF** (Li et al., 2021), an SGD-based DP MF adapted to implicit data via a binary-regression loss with the same negative sampling as DPIMF and a re-derived per-gradient sensitivity; and **EFVAE+DP-Adam** (Zhang et al., 2024), a federated VAE filter made $\varepsilon$-DP by replacing Adam with DP-Adam under the same accountant. Our variants are **DPIMF$_{str}$** (strawman, Section 3), **DPIMF$_{com}$** (complementary loss, no symmetric noise), **DPIMF$_{sym}$** (with symmetric noise), **DPIMF$_{opt}$** ($\alpha_0 = 1$), and **DPIMF$_{opt}$-IS** (with importance sampling). We set the number of sub-datasets $K = 10$; full hyper-parameters are in App. K.1.

### 6.2. Experimental Results and Analysis

#### 6.2.1. THE EFFECT OF COMPLEMENTARY LOSS

We evaluate the effectiveness of the proposed complementary loss (Section 4) by comparing the recommendation performance of DPIMF$_{str}$, DPIMF$_{com}$, DPIMF$_{sym}$, and DPIMF$_{opt}$.

We first examine its impact on reducing noise magnitude in the second-order coefficients $\tilde{\boldsymbol{B}}$ (or $\boldsymbol{B}$), using ML-10M with $\alpha_0 = 0.8$, $\varepsilon = 0.1$, and $(\beta_1, \beta_2, \beta_3) = (0.1, 0.8, 0.1)$. As shown in Figure 4(a), DPIMF$_{sym}$ exhibits the sharpest noise

---

[1]http://www.grouplens.org

[2]http://research.yahoo.com/AcademicRelations

[3]https://jmcauley.ucsd.edu/data/amazon/

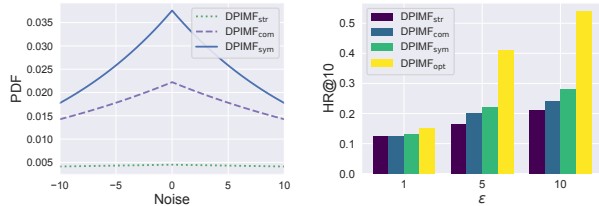

(a) ML-10M, noise distribution     (b) ML-10M, HR@10

*Figure 4.* Performance of DPIMF methods on noise distribution and HR@10.

distribution, indicating the lowest variance, while DPIMF$_{str}$ has the flattest curve and highest variance. These results confirm that the symmetric noise strategy effectively reduces noise magnitude, thereby improving recommendation accuracy.

To validate the effectiveness of complementary loss, we evaluate the HRs of DPIMF$_{str}$, DPIMF$_{com}$, and DPIMF$_{sym}$ on ML-10M, with $\alpha_0 = 0.8$ for the latter two. As shown in Figure 4(b), utility improves with increasing $\varepsilon$, consistent with DP theory. DPIMF$_{str}$ exhibits the worst performance due to high-variance noise, while DPIMF$_{sym}$ consistently outperforms DPIMF$_{com}$, achieving a 2.2% higher average HR by leveraging the symmetric noise strategy, which reduces sensitivity via upper-triangular summation. DPIMF$_{opt}$ performs best across all $\varepsilon$, as setting $\alpha_0 = 1$ eliminates perturbation on second-order terms, allowing more budget for low-variance noise and significantly enhancing utility. Results on other datasets are provided in App. K.4.

### 6.2.2. EFFECT OF IMPORTANCE SAMPLING

We study importance sampling by comparing it with uniform sampling under a fixed expected sample size. To isolate the effect of sampling on noise, we focus on the noisy linear coefficient $h_i$ in the private loss and define the corresponding mechanism $\mathcal{A}_1(\mathcal{D}) = 2 \sum_{u \in U_i} w_{ui} p_u + b$. When $\mathcal{D} = U_i$ is subsampled by the Poisson importance sampler $\mathcal{S}_W$, we evaluate the resulting variance of $\mathcal{A}_1 \circ \mathcal{S}_W$ across dimensions.

Let $m_0$ denote the target expected sample size. We compare three strategies: (i) uniform sampling, (ii) our privacy-aware importance sampling, and (iii) a utility-optimal baseline obtained by minimizing the variance of $\mathcal{A}_1(\mathcal{D})$ under the same sample-size constraint. We use 2,000 users from ML-10M with latent dimension $d = 20$ and set $m_0 \in \{20, 100\}$. We instantiate the privacy loss profile $\phi$ from the user-profile magnitude $\|p_u\|_1$; since the local profiles $P$ are fixed and stored within each silo during item-profile updates, this incurs no additional privacy budget. Fig. 5 visualizes the resulting weights $w_{ui}$. The weights produced by our method closely match the utility-optimal solution, while uniform sampling is markedly different. This alignment is structural: $\|p_u\|_1$ governs both the privacy cost (through $\phi$) and the estimation variance (through $\mathcal{A}_1$), so the privacy-aware and

utility-optimal weights nearly coincide. This suggests that, in this setting, privacy-aware importance sampling aligns well with utility optimization, and uniform sampling can be significantly suboptimal. We next show that these gains translate into improved recommendation performance.

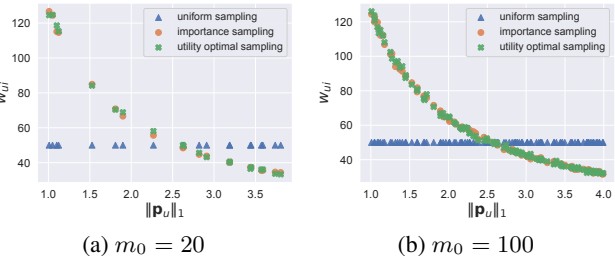

(a) $m_0 = 20$     (b) $m_0 = 100$

*Figure 5.* Comparison of sampling strategies.

### 6.2.3. COMPARISON WITH NON-ITERATIVE DP RECOMMENDERS

Differentially private recommenders fall into two regimes by how DP noise is introduced. We first compare against *non-iterative* methods—the closest match to DPIMF—which perturb the input or objective once: DPMF shares DPIMF's one-shot objective perturbation (for explicit feedback), while DPLCF and LDPICF target implicit feedback (via input perturbation). Section 6.2.4 then turns to *DP-SGD* methods, which perturb gradients at every step. To demonstrate the effectiveness of our approach, we compare DPIMF methods (i.e., DPIMF$_{opt}$ and DPIMF$_{opt}$-IS, referred to as DPIMF and DPIMF-IS) against the non-private baseline (GT) and existing DP baselines (DPMF, DPLCF, and LDPICF). Figure 6 reports HR and NDCG on ML-10M across privacy budgets $\varepsilon \in [1, 10]$; results on YahooMusic and Amazon are deferred to App. K.5.

As expected, utility improves for all private methods as $\varepsilon$ increases, aligning with the fundamental trade-off of DP. DPIMF consistently outperforms DPMF, DPLCF, and LD-PICF on all datasets, with DPIMF-IS achieving the best performance. On ML-10M, DPIMF-IS yields average HR improvements of 18.3% and 8.3% over DPLCF and LD-PICF, respectively, and NDCG improvements of 12.1% and 5.4%.

The superior performance of DPIMF methods stems from both the expressiveness of matrix factorization models and the design of low-sensitivity private losses. Compared to KNN-based methods, our strategies introduce less noise while maintaining rigorous privacy guarantees. Moreover, the advantage of DPIMF-IS becomes more pronounced under tighter privacy (e.g., $\varepsilon < 5$), validating the amplification effect described in Section 4.4.

We further evaluate DPIMF in the setting where user profiles are privatized by splitting data by items. Results on ML-10M (Figure 7) show overall accuracy improvement, with DPIMF-IS approaching GT when $\varepsilon > 5$. This is attributed

to denser user interactions (Table 1), which better offset DP noise. Similar trends are observed on other datasets (see App. K.5).

### 6.2.4. COMPARISON WITH DP-SGD RECOMMENDERS

Complementing the non-iterative DP recommenders above, we now compare against DP-SGD recommenders, which inject noise into gradients at every step, represented by SGDMF (Li et al., 2021) and EFVAE+DP-Adam (Zhang et al., 2024) (introduced in Section 6.1.3). For fairness, both baselines are evaluated under the same user-partitioned setting as DPIMF's item-profile scenario (data split by users with shared item profiles); for SGDMF this corresponds to its horizontal federated version. Both rely on gradient clipping and per-step Gaussian noise, unlike DPIMF's one-shot objective perturbation. Table 2 reports HR@10 on ML-10M across privacy budgets $\varepsilon \in \{0.5, 1, 2, 3, 5, 7, 10\}$.

*Table 2.* HR@10 on ML-10M against DP-SGD-based baselines.

| Method | $\varepsilon$=0.5 | $\varepsilon$=1 | $\varepsilon$=2 | $\varepsilon$=3 | $\varepsilon$=5 | $\varepsilon$=7 | $\varepsilon$=10 |
|---|---|---|---|---|---|---|---|
| DPIMF-IS | **0.33** | **0.35** | **0.36** | **0.37** | **0.43** | **0.50** | **0.54** |
| SGDMF | 0.15 | 0.19 | 0.25 | 0.30 | 0.37 | 0.43 | 0.48 |
| EFVAE+DP-Adam | 0.11 | 0.16 | 0.21 | 0.25 | 0.32 | 0.38 | 0.43 |

DPIMF-IS consistently outperforms both baselines across all privacy levels, and the gap widens as $\varepsilon$ decreases. At $\varepsilon = 0.5$, DPIMF-IS exceeds SGDMF and EFVAE+DP-Adam by 0.18 and 0.22 in absolute HR@10, while at $\varepsilon = 10$ the gaps narrow to 0.06 and 0.11. This pattern reflects the noise-accumulation behavior intrinsic to DP-SGD: per-step Gaussian noise compounds over many gradient updates and severely degrades utility under tight privacy, whereas DPIMF perturbs the per-item objective only once and benefits from the privacy amplification of importance sampling analyzed in Section 4.4.

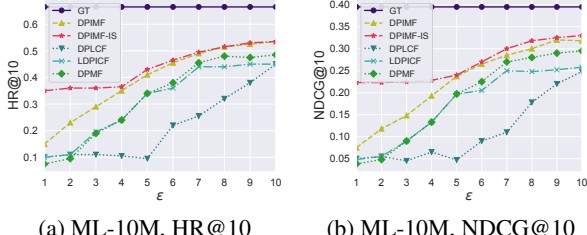

(a) ML-10M, HR@10     (b) ML-10M, NDCG@10

*Figure 6.* Results of solving the private item profile matrix.

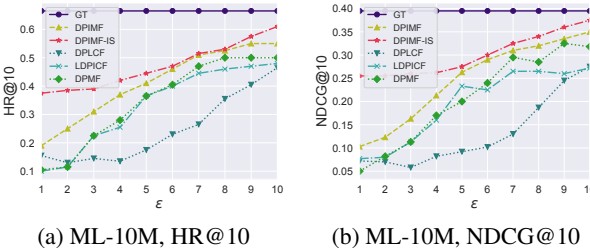

(a) ML-10M, HR@10     (b) ML-10M, NDCG@10

*Figure 7.* Results of solving the private user profile matrix.

## 7. Related Work

Differential privacy has been widely applied to recommender systems. McSherry et al. (McSherry & Mironov, 2009) first brought DP to collaborative filtering via item-covariance perturbation; later work added noise to KNN similarities or predictions (Zhu et al., 2014; Wang & Tang, 2017), applied stochastic gradient Langevin dynamics for private item profiles (Liu et al., 2015; Wang et al., 2015; Jiang et al., 2019), and extended DP to graph neural networks for link prediction (Ran et al., 2024).

Jain et al. (Jain et al., 2018) introduced joint differential privacy (JDP) via a private Frank–Wolfe algorithm, later extended to JDP-preserving matrix factorization through alternating least squares (Chien et al., 2021). Berlioz et al. (Berlioz et al., 2015) classified MF mechanisms into input, in-process, and output perturbation, and Hua et al. (Hua et al., 2015) proposed objective perturbation for DPMF. Recently, federated collaborative filters such as SGDMF (Li et al., 2021) adopt the DP-SGD paradigm (Fu et al., 2024)—per-step gradient clipping and Gaussian noise for $(\varepsilon, \delta)$-DP across silos (Cai et al., 2024)—whereas others such as the federated VAE EFVAE (Zhang et al., 2024) are not private by default; the local-DP-versus-federated-learning trade-off (Zheng et al., 2020) and membership inference (Bai et al., 2024) have likewise been studied. In contrast to the rich DP literature on explicit feedback, only a few KNN-based methods address implicit feedback (Guo et al., 2019; Gao et al., 2020), despite its strong privacy risks (Weinsberg et al., 2012; Narayanan & Shmatikov, 2008), and they generalize poorly on sparse, high-dimensional data (Rendle et al., 2022). In contrast, DPIMF brings objective perturbation to implicit-feedback MF in the cross-silo setting, attaining pure $\varepsilon$-DP rather than the $(\varepsilon, \delta)$-DP of gradient-based federated filters.

## 8. Conclusion

We study differentially private implicit matrix factorization for cross-silo recommendation, a setting in which one-class observations and aggressive negative sampling make standard objective perturbation prohibitively noisy. We propose DPIMF, which combines a complementary loss that offsets item-dependent sensitivity, spectral truncation that keeps the privatized objective well-posed, and a privacy-aware importance sampler that further amplifies privacy with minimal utility cost. Formal excess-risk bounds characterize when each design element pays off, and experiments on three real-world benchmarks confirm that DPIMF achieves a markedly better privacy–utility trade-off than both objective-perturbation and DP-SGD-style baselines, with the advantage most pronounced under tight privacy budgets. Richer interaction signals and heterogeneous cross-silo trust are natural future directions.

## Acknowledgements

This work was supported by the National Natural Science Foundation of China (Grant No: 62372122), the Research Grants Council (Grant No: 25207224, 15208825, and C2003-23Y), and the Innovation and Technology Fund (Grant No: GHP/392/22GD), Hong Kong SAR, China.

## Impact Statement

This work aims to advance the field of machine learning by developing a differentially private framework for cross-silo recommendation from implicit feedback. By enabling collaborative model training without sharing raw user interaction data, the proposed DPIMF framework has the potential to improve privacy protection in real-world recommender systems deployed across multiple organizations.

From a societal perspective, stronger privacy guarantees in recommendation systems may help mitigate risks of unintended information leakage, such as user profiling or inference attacks, thereby contributing to more responsible data use. This is particularly relevant for applications involving sensitive behavioral signals, including online browsing, consumption, or engagement patterns.

At the same time, as with many privacy-preserving learning techniques, the proposed methods do not eliminate all possible risks. Differential privacy provides formal guarantees under specific threat models and assumptions, and misuse or misinterpretation of these guarantees could lead to a false sense of security if deployed improperly. Moreover, improved recommendation accuracy, even under privacy constraints, may still reinforce existing biases present in the underlying data, an issue that is orthogonal to the scope of this work.

Overall, we believe the primary impact of this work is methodological, contributing tools and insights for building privacy-aware recommender systems. Any broader societal consequences are consistent with those commonly associated with advances in privacy-preserving machine learning, and we do not foresee significant negative impacts beyond those already well studied in the field.

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

# A. Implementation of Full-fledged DPIMF

In this section, we show the implementation details of the full-fledged DPIMF. We first provide the process for solving the private item profile matrix, as shown in Algorithm 2. Then we show how to extend the algorithm to solve the private user profile matrix.

It starts by initializing the profile matrix and sensitivities (Lines 1-2). Here, we uniformly sample random values between 0 and 1 by following the approach used in existing literature (Berlioz et al., 2015). We then pre-compute the global Gramian $G_U = \sum_{u \in U} p_u \otimes p_u$ (Line 3). Because flipping a single interaction never changes the set of all user profiles, $G_U$ is item-independent and insensitive; hence the term $\alpha_0 G_U$ in the cancellation of Eq. (12) carries zero sensitivity and is computed once, while only the item-local term $(1 - \alpha_0) G_{U_i}$ is sensitive — this is the offsetting effect. Next, the privacy loss profile is determined and the per-user importance weights $\{w_u\}_{u \in U}$ are obtained by solving the optimization problem detailed in Section 4.4 (Lines 4-5). For each item, we apply Poisson importance sampling to the observed interactions — retaining each $u \in U_i$ with probability $1/w_u$ — and form the sensitive data-dependent coefficients $G_{U_i}$ and $g_{U_i}$ over the retained set, reweighted by $w_u$ (Lines 6-8). The reweighting debiases the subsample, so in expectation these coefficients recover the deterministic complementary loss, while the subsampling amplifies privacy (App. F). Next, Laplacian noise is sampled and symmetrized (Lines 9-10) and combined with these coefficients to form the private quadratic, linear, and regularization terms $M_i, h_i$, and $\lambda_i$ (Line 11), exactly as in Eq. (14). Spectral truncation is then applied to $M_i + \lambda_i E$, where $E$ is the $d \times d$ identity, to construct the final private loss (Lines 12-13). The optimal private profile vector is solved within a convex set (Line 14).

Algorithm 2 solves the item profile matrix. We now discuss how to extend it to the user profile matrix. When solving a user's profile vector $p_u$, we focus on the items that $u$ has interacted with, denoted as $I_u$. In this case, the input $P$ and $\{U_i \mid i \in I\}$ should be replaced by $Q$ and $\{I_u \mid u \in U\}$, respectively. The weight-solving steps (Lines 4-5) are unchanged, since $W$ is obtained from an independent problem; by Assumption (v) the relevant marginal for the user-profile update is the per-item weight $w_i$. To build the private loss of $p_u$, we replace $U_i \rightarrow I_u, \tilde{U}_i \rightarrow \tilde{I}_u, U \rightarrow I$, $p_u \rightarrow q_i$, and the per-user weight $w_u \rightarrow$ per-item weight $w_i$ throughout the global Gramian and the per-item steps (Lines 3 and 6-13). Then we can derive $\bar{p}_u$ using the same solution as for $\bar{q}_i$ (Line 14).

**Algorithm 2** Full-fledged DPIMF (Solving Private Item Profile Matrix)

**Input:** The local user profile matrix $P$, the user set $U$, the item set $I$, the set of users interacted with each item $\{U_i \mid i \in I\}, \alpha_0$, the regularization weight $\lambda$, the target privacy budget $\varepsilon'$, and the per-term budgets $\varepsilon_1, \varepsilon_2, \varepsilon_3$.

**Output:** $\bar{Q}$.

1: Randomly initialize $\bar{Q}$ such that each element $\bar{q}_{ij}$ is drawn from $[0, 1]$.
2: Compute sensitivities $\Delta_1, \Delta_2, \Delta_3$ according to Eq. (13) and App. L.
3: Pre-compute the global Gramian $G_U \leftarrow \sum_{u \in U} p_u \otimes p_u$.
   // Solve the weights for importance sampling
4: Determine the privacy loss profile $\phi$.
5: Solve the per-user weights $\{w_u\}_{u \in U}$ for target budget $\varepsilon'$ via Problem 1 (Algorithm 3).
6: **for** each $i \in I$ **do**
   // Importance-sample, then build the private complementary loss
7:   Draw a Poisson subsample $\hat{U}_i \subseteq U_i$, retaining each $u$ independently with probability $1/w_u$.
8:   Compute $G_{U_i} \leftarrow \sum_{u \in \hat{U}_i} w_u (p_u \otimes p_u)$.
9:   Compute $g_{U_i} \leftarrow \sum_{u \in \hat{U}_i} w_u p_u$.
10:  Sample $b \sim \text{Lap}\left(\frac{\Delta_1}{\varepsilon_1}\right)^d, B \sim \text{Lap}\left(\frac{\Delta_2}{\varepsilon_2}\right)^{d \times d}, \eta \sim \text{Lap}\left(\frac{\Delta_3}{\varepsilon_3}\right)$.
11:  Compute symmetric noise $\tilde{B} \leftarrow \text{triu}(B) + \text{tril}_{-1}(B^\top)$.
12:  Form the private coefficients $M_i \leftarrow \alpha_0 G_U + (1 - \alpha_0) G_{U_i} + \tilde{B}, h_i \leftarrow 2g_{U_i} + b$, and $\lambda_i \leftarrow \lambda \left(|U_i| + \alpha_0 |\tilde{U}_i| + \eta\right)$.
13:  Perform spectral truncation on $(M_i + \lambda_i E)$ to obtain a positive definite matrix $M'$.
14:  Derive the private loss $\bar{L}(q_i) \leftarrow q_i^\top M' q_i - q_i^\top h_i$.
15:  Solve $\bar{q}_i \leftarrow \arg\min_{q_i: \|q_i\|_2 \leq \sqrt{1/\lambda}} \bar{L}(q_i)$.
16: **end for**
17: **return** $\bar{Q}$.

# B. Proof of the unique solution of Problem 1

*Problem* 1 (Privacy-utility optimal sampling). For a privacy loss profile $\phi: [1, \infty) \times (U \times I) \rightarrow \mathbb{R}$, a target privacy guarantee $\varepsilon'$, and the set of all user-item pairs $U \times I$, we define the privacy-optimal sampling problem as

$$\underset{W \in \mathbb{R}^{|U| \times |I|}}{\arg\min} \quad \sum_{(u,i) \in U \times I} \frac{1}{w_{ui}}$$

$$\text{s.t.} \quad \ln\left(1 + \frac{1}{w_{ui}}\left(e^{\phi(w_{ui}, u, i)} - 1\right)\right) \leq \varepsilon', \quad \forall u, i,$$

$$w_{ui} \geq 1, \qquad \forall u, i.$$

**Assumptions on $\phi$.** For each $(u, i) \in U \times I$ we assume: (i) $\phi(\cdot, u, i)$ is differentiable on $[1, \infty)$ and depends only on $(w, u, i)$—not on the private interactions—so that $W^*$ is data-independent; (ii) the amplified loss $\ln(1 + \frac{1}{w}(e^{\phi(w,u,i)} - 1))$ is monotone in $w$, so the constraint binds at a single weight; (iii) $\varepsilon' \geq \phi(1, u, i)$, so that $w_{ui} = 1$ is feasible; (iv) the per-pair objective is $\mu_{ui}$-strongly convex with $\mu_{ui} > 0$; and (v) for the item-profile update, $\phi$ is a per-

user profile, i.e., $\phi(w, u, i) = \phi(w, u)$ does not depend on $i$—a natural choice here, as a record's sensitivity is governed by the user profile $\boldsymbol{p}_u$ rather than by the item it belongs to. With the bounded, non-empty feasible region, (iii)–(v) yield the unique solution below. By (v), the per-pair subproblems for a fixed user are identical, so the optimal weights satisfy $w_{ui} = w_u$ (i.e., $\boldsymbol{W}$ is constant across items), and Problem 1 reduces to one univariate root per user. The per-user weight defines a single keep-probability $1/w_u$ for every interaction of user $u$; importance sampling therefore acts only on the sensitive item-local term, while the global aggregate $\boldsymbol{G_U} = \sum_{u \in \boldsymbol{U}} \boldsymbol{p}_u \otimes \boldsymbol{p}_u$ is computed deterministically over all profiles and is insensitive to any single interaction, so the cancellation in Eq. (12) (the offsetting effect) is preserved (see App. A). (Solving the user profile matrix is symmetric, with a per-item weight $w_{ui} = w_i$.)

*Theorem* 5. Let $\varepsilon' \geq \phi(1, u)$ for all $u \in \boldsymbol{U}$ and $\phi(w, u) > \ln(1 + w(e^{\varepsilon'} - 1))$ for all $w \geq v_u$ where $v_u \geq 1$. Problem 1 has a unique solution $\boldsymbol{W}^*$.

*Proof:* Without loss of generality, we consider each $w_u$ independently. Since the objective $\sum_u 1/w_u$ is separable and strictly decreasing in each $w_u$, minimizing it is equivalent to maximizing each weight subject to the per-user constraint:

$$\operatorname*{arg\,max}_{\{w_u\}_{u \in \boldsymbol{U}}} \quad \sum_{u \in \boldsymbol{U}} w_u$$
$$\text{s.t.} \quad \ln\left(1 + \frac{1}{w_u}\left(e^{\phi(w_u, u)} - 1\right)\right) \leq \varepsilon', \quad \forall u,$$
$$w_u \geq 1, \qquad\qquad\qquad \forall u.$$
$$(18a)$$

According to the setting we concerned, the feasible region is bounded and not empty. Since the objective is strictly monotonic, the solution must be unique. ∎

## C. Algorithm for solving Problem 1

Problem 1 can be solved by the bisection-based convex optimizer described in Algorithm 3; since $\phi$ is per-user (Assumption (v)), it suffices to solve one univariate equation per user. Here $\text{BISECT}(g, I, \rho)$ returns a root of $g$ within the interval $I$ to tolerance $\rho$, converging in $O(\log(1/\rho))$ steps.

Here, strong convexity is defined as follows:

*Definition* 2. (Strong Convexity). Let $\mu > 0$. A differentiable function $f : \mathbb{R}^d \to \mathbb{R}$ is $\mu$-strongly convex if for all $u, v \in \mathbb{R}^d$

$$f(v) \geq f(u) + \nabla f(u)^\top (v - u) + \frac{\mu}{2}\|v - u\|_2^2 \quad (19)$$

## D. Detailed Time Complexity Analysis

We detail the per-round cost of the full-fledged DPIMF in Section 4 (Algorithm 2) for updating the item profile matrix; the user side is symmetric, supporting the summary in Sec-

---

**Algorithm 3** Privacy-Utility Optimal Sampling

**Input:** User set $\boldsymbol{U}$; target privacy budget $\varepsilon'$; per-user privacy loss profile $\phi$; strong convexity constants $\mu_u$ for $u \in \boldsymbol{U}$; tolerance $\rho$.
**Output:** Optimal per-user sampling weights $W^* = \{w_u\}_{u \in \boldsymbol{U}}$ (set $w_{ui} = w_u$).
1: **for all** $u \in \boldsymbol{U}$ **do**
2:    Compute

$$g_u(w) = \frac{\exp(\phi(w, u)) - 1}{w} - \left(\exp(\varepsilon') - 1\right).$$

3:    Compute the upper bound

$$v_u \leftarrow \frac{2\left(\exp(\varepsilon') - \frac{\partial \phi(1, u)}{\partial w} \exp(\phi(1, u)) + 1\right)}{\mu_u} + 1.$$

4:    **if** $\phi(1, u) = \varepsilon'$ **or** $\phi(1, u) < 0$ **then**
5:       $w_u \leftarrow \text{BISECT}\left(g_u, (1, v_u], \rho\right)$
6:    **else**
7:       $w_u \leftarrow \text{BISECT}\left(g_u, [1, v_u], \rho\right)$
8:    **end if**
9: **end for**
10: **return** $W^*$

---

tion 5.3. Let $n = |\boldsymbol{U}|$, $m = |\boldsymbol{I}|$, and $|\mathcal{S}| = \sum_i |\boldsymbol{U}_i|$ denote the numbers of users, items, and observed interactions, and let $d$ be the latent dimension.

**Preprocessing.** The shared Gramian $\boldsymbol{G_U} = \sum_{u \in \boldsymbol{U}} \boldsymbol{p}_u \otimes \boldsymbol{p}_u$ is computed once in $O(nd^2)$ and reused across all items. The importance weights $\boldsymbol{W}$ are obtained by Algorithm 3; since the weights are per-user ($w_{ui} = w_u$), it solves only $|\boldsymbol{U}| = n$ univariate equations by bisection (one per user), and with tolerance $\rho$ each call costs $O(\log(1/\rho))$, giving $O(n \log(1/\rho))$ in total.

**Per-item update.** For each item $i$, the cost decomposes as: (i) drawing the Poisson subsample $\widehat{\boldsymbol{U}}_i \subseteq \boldsymbol{U}_i$ and forming the reweighted per-item Gramian $\boldsymbol{G}_{\boldsymbol{U}_i}$ and linear term $\boldsymbol{g}_{\boldsymbol{U}_i}$ takes $O(|\boldsymbol{U}_i| d^2)$; (ii) assembling the quadratic coefficient via Eq. (12) and drawing the Laplace noises $\tilde{\boldsymbol{B}}, \boldsymbol{b}, \eta$ costs $O(d^2)$; (iii) spectral truncation requires one eigendecomposition of a $d \times d$ matrix in $O(d^3)$ (when $\alpha_0 = 1$, $\Delta_2 = \Delta_3 = 0$ by Eq. (13), so $\boldsymbol{M}$ is already positive semidefinite and this step is skipped); (iv) solving the constrained quadratic program over $\{\boldsymbol{q}_i : \|\boldsymbol{q}_i\|_2 \leq \sqrt{1/\lambda}\}$ reuses the eigenbasis from (iii) and reduces to a one-dimensional root search, contributing only $O(d^2)$. The dominant per-item cost is therefore $O(|\boldsymbol{U}_i| d^2 + d^3)$.

**Total cost.** Summing the per-item cost over all items and adding the preprocessing yields

$$O\left(|\mathcal{S}| d^2 + md^3 + nd^2 + n \log(1/\rho)\right).$$

A symmetric pass over users swaps the roles of $n$ and $m$, so one full round of alternating updates costs

$$O\left(|\mathcal{S}| d^2 + (n + m) d^3 + n \log(1/\rho)\right).$$

**Parallelism.** Each item update in Algorithm 2 (lines 6–

14) depends only on the shared Gramian $G_U$ and on item $i$'s own interaction column $U_i$, so the per-item updates are mutually independent and the item sweep parallelizes across the $m$ items (symmetrically, the user sweep across the $n$ users). In the cross-silo setting, each party processes only its local partition, distributing this workload naturally.

## E. Technical Lemmas

*Lemma* 1. Given the complementary loss $L(q_i)$ of Eq. (10) and the original loss $L^O(q_i)$ of Eq. (6), let $q_i^* = \arg\min L(q_i)$ and $q_i' = \arg\min L^O(q_i)$, then $\|q_i^*\|_2 \leq \sqrt{\frac{1}{\lambda}}$, and $\|q_i'\|_2 \leq \sqrt{\frac{1}{\lambda}}$.

*Proof:* Write $L(q_i) = L_I(q_i) + R(q_i)$, where $L_I(q_i)$ collects the squared-error terms and $R(q_i) = \lambda(|U_i| + \alpha_0|\tilde{U}_i|)\|q_i\|_2^2$ is the regularizer. Since $L_I(q_i) \geq 0$, we have $L(q_i) \geq R(q_i)$. The optimal solution $q_i^*$ satisfies $L(q_i^*) \leq L(q_i)$ for any $q_i$. Taking $q_i = 0$ gives $L(q_i^*) \leq L(0) = |U_i|$. Since $\alpha_0 \in [0, 1]$, we have

$$\lambda\left(|U_i| + \alpha_0|\tilde{U}_i|\right)\|q_i^*\|_2^2 \leq |U_i|$$

$$\|q_i^*\|_2 \leq \sqrt{\frac{|U_i|}{\lambda\left(|U_i| + \alpha_0|\tilde{U}_i|\right)}}$$

$$\leq \sqrt{\frac{1}{\lambda}}$$

Similarly, we can derive $\|q_i'\|_2 \leq \sqrt{\frac{1}{\lambda}}$, which completes the proof. ∎

*Lemma* 2. Let $x = (x_1, x_2, \cdots, x_d)^T$, where $x_j \sim \text{Lap}\left(\frac{\Delta}{\varepsilon}\right)$, $j \in \{1, 2, \cdots, d\}$. Then $\mathbb{E}\left[\|x\|_2\right] \leq \frac{\sqrt{2d}\Delta}{\varepsilon}$.

*Proof:* According to the Jensen's inequality, we have

$$\mathbb{E}\left[\|x\|_2\right] = \mathbb{E}\left[\sqrt{\sum_j x_j^2}\right] \leq \sqrt{\mathbb{E}\left[\sum_j x_j^2\right]}.$$

Since the expectation of the laplace distribution is zero, we have

$$\mathbb{E}\left[\sum_j x_j^2\right] = \sum_j \mathbb{E}\left[x_j^2\right] = \sum_j 2\frac{\Delta^2}{\varepsilon^2} = \frac{2d\Delta^2}{\varepsilon^2}.$$

Therefore, we have $\mathbb{E}\left[\|x\|_2\right] \leq \frac{\sqrt{2d}\Delta}{\varepsilon}$. ∎

## F. Proof of Theorem 1

*Proof:* Let $\mathcal{M}$ be the base mechanism with privacy loss profile $\phi$: for weighted datasets differing in a single record $z = (u, i)$ included at weight $w$, and any measurable set $A$, $\Pr[\mathcal{M}(D) \in A] \leq e^{\phi(w,u,i)}\Pr[\mathcal{M}(D') \in A]$. The Poisson importance sampler $\mathcal{S}_W$ retains each record $(u, i)$ independently with probability $1/w_{ui}$, and $\widehat{\mathcal{M}} = \mathcal{M} \circ \mathcal{S}_W$.

Fix neighboring datasets $D = D' \cup \{z\}$ with $z = (u, i)$, a measurable set $A$, and write $q = 1/w_{ui}$. Let $P(Z) = \Pr[\widehat{\mathcal{M}}(D) \in A \mid \mathcal{S}_W(D) = Z]$ and $P'(Z) = \Pr[\widehat{\mathcal{M}}(D') \in A \mid \mathcal{S}_W(D') = Z]$. Conditioning on whether $z$ is sampled,

$$\begin{aligned}\Pr[\widehat{\mathcal{M}}(D) \in A] = {} &q\,\mathbb{E}[P(\mathcal{S}_W(D)) \mid z \in \mathcal{S}_W(D)] \\ &+ (1-q)\,\mathbb{E}[P(\mathcal{S}_W(D)) \mid z \notin \mathcal{S}_W(D)].\end{aligned}$$
(20)

The selection events are independent, so conditioned on $z \notin \mathcal{S}_W(D)$ the samples $\mathcal{S}_W(D)$ and $\mathcal{S}_W(D')$ are identically distributed, giving

$$\mathbb{E}[P(\mathcal{S}_W(D)) \mid z \notin \mathcal{S}_W(D)] = \mathbb{E}[P'(\mathcal{S}_W(D'))]. \quad (21)$$

Conditioned on $z \in \mathcal{S}_W(D)$, the sets $\mathcal{S}_W(D)$ and $\mathcal{S}_W(D) \setminus \{z\}$ are neighboring and $\mathcal{S}_W(D) \setminus \{z\}$ is distributed as $\mathcal{S}_W(D')$; applying the profile of $\mathcal{M}$ at weight $w_{ui}$,

$$\mathbb{E}[P(\mathcal{S}_W(D)) \mid z \in \mathcal{S}_W(D)] \leq e^{\phi(w_{ui}, u, i)}\,\mathbb{E}[P'(\mathcal{S}_W(D'))]. \quad (22)$$

Substituting (21) and (22) into (20),

$$\Pr[\widehat{\mathcal{M}}(D) \in A] \leq \left(1 + q(e^{\phi(w_{ui}, u, i)} - 1)\right)\Pr[\widehat{\mathcal{M}}(D') \in A].$$

With $q = 1/w_{ui}$, the amplified privacy loss of record $(u, i)$ is $\ln\left(1 + \frac{1}{w_{ui}}(e^{\phi(w_{ui}, u, i)} - 1)\right)$; a symmetric argument gives the matching lower bound. Taking the maximum over $(u, i) \in U \times I$ yields $\varepsilon'$-DP with $\varepsilon' = \max_{(u,i) \in U \times I} \ln\left(1 + \frac{1}{w_{ui}}(e^{\phi(w_{ui}, u, i)} - 1)\right)$. The argument follows the standard privacy amplification by subsampling with $\delta = 0$ (Steinke, 2025). ∎

## G. Proof of Theorem 2

*Proof:* According to our design logic, spectral truncation and importance sampling do not require further access to the input database. Thus, based on the post-processing theorem, demonstrating that solving $\bar{q}_i$ for the private loss $\bar{l}(q_i)$ in Eq. (14) satisfies $\varepsilon$-DP for any arbitrary item $i$ is sufficient to establish that Algorithm 2 satisfies DP. Let $T \in \mathbb{R}^{d \times d}, t \in \mathbb{R}^d$, and $\tau \in \mathbb{R}$ be the perturbed coefficients of the second and first-order terms in $\bar{l}_I(q_i)$, and the perturbed weight of the regularization term $\bar{R}(q_i)$ of Eq. (14), respectively. Define dataset $D_i$ that records both the positive and non-positive users for $i$, then its neighboring dataset $D_i'$ is obtained from $D_i$ by moving one user from the positive set to the non-positive set (and vice versa). For a privacy

budget $\varepsilon$ divided into $\varepsilon_1, \varepsilon_2, \varepsilon_3$, we have

$$\frac{\Pr\{\bar{l}(\boldsymbol{q}_i) \mid D_i\}}{\Pr\{\bar{l}(\boldsymbol{q}_i) \mid D_i'\}}$$

$$= \frac{\Pr\{\sum_{u \in \boldsymbol{U}_i} \boldsymbol{p}_u \otimes \boldsymbol{p}_u + \alpha_0 \sum_{u \in \tilde{\boldsymbol{U}}_i} \boldsymbol{p}_u \otimes \boldsymbol{p}_u + \tilde{\boldsymbol{B}} = \boldsymbol{T}\}}{\Pr\{\sum_{u \in \boldsymbol{U}_i'} \boldsymbol{p}_u \otimes \boldsymbol{p}_u + \alpha_0 \sum_{u \in \tilde{\boldsymbol{U}}_i'} \boldsymbol{p}_u \otimes \boldsymbol{p}_u + \tilde{\boldsymbol{B}}' = \boldsymbol{T}\}}$$

$$\cdot \frac{\Pr\{2 \sum_{u \in \boldsymbol{U}_i} \boldsymbol{p}_u + \boldsymbol{b} = \boldsymbol{t}\} \cdot \Pr\{|\boldsymbol{U}_i| + \alpha_0|\tilde{\boldsymbol{U}}_i| + \eta = \tau\}}{\Pr\{2 \sum_{u \in \boldsymbol{U}_i'} \boldsymbol{p}_u + \boldsymbol{b}' = \boldsymbol{t}\} \cdot \Pr\{|\boldsymbol{U}_i'| + \alpha_0|\tilde{\boldsymbol{U}}_i'| + \eta' = \tau\}}$$

$$= \frac{\prod_{j \leq l} \Pr\left\{\tilde{B}_{jl} = T_{jl} - (\boldsymbol{G}_{\boldsymbol{U}_i} + \alpha_0 \boldsymbol{G}_{\tilde{\boldsymbol{U}}_i})_{jl}\right\}}{\prod_{j \leq l} \Pr\left\{\begin{array}{c}\tilde{B}_{jl}' = T_{jl} - (\boldsymbol{G}_{\boldsymbol{U}_i} + \alpha_0 \boldsymbol{G}_{\tilde{\boldsymbol{U}}_i})_{jl} - \\ (1-\alpha_0)(\boldsymbol{p}_v \otimes \boldsymbol{p}_v)_{jl}\end{array}\right\}}$$

$$\cdot \frac{\prod_{j=1}^d \Pr\left\{b_j = t_j - (2\sum_{u \in \boldsymbol{U}_i'} \boldsymbol{p}_u)_j\right\}}{\prod_{j=1}^d \Pr\left\{b_j = t_j - (2\sum_{u \in \boldsymbol{U}_i'} \boldsymbol{p}_u)_j - 2p_{vj}\right\}}$$

$$\cdot \frac{\Pr\left\{\eta = \tau - |\boldsymbol{U}_i| - \alpha_0|\tilde{\boldsymbol{U}}_i|\right\}}{\Pr\left\{\eta = \tau - |\boldsymbol{U}_i| - \alpha_0|\tilde{\boldsymbol{U}}_i| - (1-\alpha_0)\right\}}$$

$$\leq \exp\left(\frac{\varepsilon_2 \max_{u \in \boldsymbol{U}}((1-\alpha_0)\|\boldsymbol{p}_u \otimes \boldsymbol{p}_u\|_{\text{triu}})}{\Delta_2}\right)$$

$$\cdot \exp\left(\frac{\varepsilon_1 \max_{u \in \boldsymbol{U}}(2\|\boldsymbol{p}_u\|_1)}{\Delta_1}\right) \exp\left(\frac{(1-\alpha_0)\varepsilon_3}{\Delta_3}\right)$$

$$= \exp(\varepsilon_1 + \varepsilon_2 + \varepsilon_3) = \exp(\varepsilon),$$

where $\boldsymbol{G}_{\boldsymbol{U}_i} = \sum_{u \in \boldsymbol{U}_i} \boldsymbol{p}_u \otimes \boldsymbol{p}_u$, $\boldsymbol{G}_{\tilde{\boldsymbol{U}}_i} = \sum_{u \in \tilde{\boldsymbol{U}}_i} \boldsymbol{p}_u \otimes \boldsymbol{p}_u$. Thus solving $\bar{\boldsymbol{q}}_i$ satisfies $\varepsilon$-DP. Since the interactions of each item are disjoint, according to the parallel composition property of DP (Dwork et al., 2014), we conclude that DPIMF satisfies $\varepsilon$-DP. ∎

## H. Proof of Theorem 3

*Proof:* Recall that $\boldsymbol{q}_i^* = \arg\min L(\boldsymbol{q}_i)$ and $\boldsymbol{q}_i' = \arg\min L^{\text{O}}(\boldsymbol{q}_i)$, where $L$ and $L^{\text{O}}$ are the complementary loss of Eq. (10) and the original loss of Eq. (6), respectively. Let $\boldsymbol{G}_{\boldsymbol{U}_i} = \sum_{u \in \boldsymbol{U}_i} \boldsymbol{p}_u \otimes \boldsymbol{p}_u$. The two losses differ only on the $\boldsymbol{U}_i$ block:

$$L^{\text{O}}(\boldsymbol{q}_i) - L(\boldsymbol{q}_i) = \alpha_0 \boldsymbol{q}_i^\top \boldsymbol{G}_{\boldsymbol{U}_i} \boldsymbol{q}_i + \lambda\alpha_0|\boldsymbol{U}_i| \|\boldsymbol{q}_i\|_2^2 \geq 0.$$

Hence $L(\boldsymbol{q}_i) \leq L^{\text{O}}(\boldsymbol{q}_i)$ for every $\boldsymbol{q}_i$, so $L(\boldsymbol{q}_i^*) \leq L^{\text{O}}(\boldsymbol{q}_i')$ and the left-hand side of Eq. (15) is bounded by its absolute value. Using that $\boldsymbol{q}_i'$ minimizes $L^{\text{O}}$,

$$\left|L(\boldsymbol{q}_i^*) - L^{\text{O}}(\boldsymbol{q}_i')\right| = L^{\text{O}}(\boldsymbol{q}_i') - L(\boldsymbol{q}_i^*) \leq L^{\text{O}}(\boldsymbol{q}_i^*) - L(\boldsymbol{q}_i^*)$$
$$= \alpha_0 \boldsymbol{q}_i^{*\top} \boldsymbol{G}_{\boldsymbol{U}_i} \boldsymbol{q}_i^* + \lambda\alpha_0|\boldsymbol{U}_i| \|\boldsymbol{q}_i^*\|_2^2.$$

By Lemma 1, $\|\boldsymbol{q}_i^*\|_2 \leq N$; together with $\|\boldsymbol{p}_u\|_2^2 \leq dc^2$ and the Cauchy–Schwarz inequality,

$$\boldsymbol{q}_i^{*\top} \boldsymbol{G}_{\boldsymbol{U}_i} \boldsymbol{q}_i^* = \sum_{u \in \boldsymbol{U}_i} (\boldsymbol{p}_u^\top \boldsymbol{q}_i^*)^2$$
$$\leq \sum_{u \in \boldsymbol{U}_i} \|\boldsymbol{p}_u\|_2^2 \|\boldsymbol{q}_i^*\|_2^2 \leq |\boldsymbol{U}_i| dc^2 N^2.$$

Therefore

$$L(\boldsymbol{q}_i^*) - L^{\text{O}}(\boldsymbol{q}_i') \leq \left|L(\boldsymbol{q}_i^*) - L^{\text{O}}(\boldsymbol{q}_i')\right|$$
$$\leq \alpha_0|\boldsymbol{U}_i|dc^2 N^2 + \lambda\alpha_0|\boldsymbol{U}_i|N^2$$
$$\leq \alpha_0|\boldsymbol{U}_i|N^2(\lambda + 2dc^2),$$

which establishes Eq. (15). ∎

## I. Proof of Theorem 4

*Proof:* For the ease of readability, we repeat the private loss function $\bar{l}(\boldsymbol{q}_i)$ under $\mathcal{M}_1, \mathcal{M}_2, \mathcal{M}_3, \mathcal{M}_4$ as follows:

$$\bar{l}^{(1)}(\boldsymbol{q}_i) = \boldsymbol{q}_i^T (\sum_{u \in \boldsymbol{U}_i} \boldsymbol{p}_u \otimes \boldsymbol{p}_u + \alpha_0 \sum_{u \in \boldsymbol{U}} \boldsymbol{p}_u \otimes \boldsymbol{p}_u + \boldsymbol{B})\boldsymbol{q}_i - $$
$$\boldsymbol{q}_i^T (2 \sum_{u \in \boldsymbol{U}_i} \boldsymbol{p}_u + \boldsymbol{b}) + \lambda (|\boldsymbol{U}_i| + \alpha_0|\boldsymbol{U}| + \eta) \|\boldsymbol{q}_i\|_2^2,$$

$$\bar{l}^{(2)}(\boldsymbol{q}_i) = \boldsymbol{q}_i^T [\sum_{u \in \boldsymbol{U}_i} \boldsymbol{p}_u \otimes \boldsymbol{p}_u + \alpha_0 \sum_{u \in \tilde{\boldsymbol{U}}_i} \boldsymbol{p}_u \otimes \boldsymbol{p}_u + \boldsymbol{B} + $$
$$\lambda(|\boldsymbol{U}_i| + \alpha_0|\tilde{\boldsymbol{U}}_i| + \eta)\boldsymbol{E}]\boldsymbol{q}_i - \boldsymbol{q}_i^T (2 \sum_{u \in \boldsymbol{U}_i} \boldsymbol{p}_u + \boldsymbol{b}),$$

$$\bar{l}^{(3)}(\boldsymbol{q}_i) = \boldsymbol{q}_i^T [\sum_{u \in \boldsymbol{U}_i} \boldsymbol{p}_u \otimes \boldsymbol{p}_u + \alpha_0 \sum_{u \in \tilde{\boldsymbol{U}}_i} \boldsymbol{p}_u \otimes \boldsymbol{p}_u + \tilde{\boldsymbol{B}} + $$
$$\lambda(|\boldsymbol{U}_i| + \alpha_0|\tilde{\boldsymbol{U}}_i| + \eta)\boldsymbol{E}]\boldsymbol{q}_i - \boldsymbol{q}_i^T (2 \sum_{u \in \boldsymbol{U}_i} \boldsymbol{p}_u + \boldsymbol{b}),$$

$$\bar{l}^{(4)}(\boldsymbol{q}_i) = \boldsymbol{q}_i^T \boldsymbol{M}' \boldsymbol{q}_i - \boldsymbol{q}_i^T \boldsymbol{h}_i$$

Where $\boldsymbol{M}'$ and $\boldsymbol{h}_i$ are defined as in Algorithm 2. For an arbitrary minimizer $\bar{\boldsymbol{q}}_i^{(k)}$ of private loss $\bar{l}^{(k)}(\boldsymbol{q}_i)$ over the convex set $C = \{\boldsymbol{q}_i \mid \|\boldsymbol{q}_i\|_2 \leq N\}$, $k = 1, 2, 3$, we have

$$L(\bar{\boldsymbol{q}}_i) - L(\boldsymbol{q}_i^*) = L(\bar{\boldsymbol{q}}_i) - \bar{l}(\boldsymbol{q}_i^*) + \bar{l}(\boldsymbol{q}_i^*) - \bar{l}(\bar{\boldsymbol{q}}_i) + \bar{l}(\bar{\boldsymbol{q}}_i) - L(\boldsymbol{q}_i^*)$$
$$\leq \bar{l}(\boldsymbol{q}_i^*) - L(\boldsymbol{q}_i^*) + L(\bar{\boldsymbol{q}}_i) - \bar{l}(\bar{\boldsymbol{q}}_i)$$
$$= (\boldsymbol{q}_i^{*T} \hat{\boldsymbol{B}} \boldsymbol{q}_i^* - 2\boldsymbol{q}_i^{*T} \boldsymbol{b} + \lambda\eta \|\boldsymbol{q}_i^*\|_2^2) + $$
$$(-\bar{\boldsymbol{q}}_i^T \hat{\boldsymbol{B}} \bar{\boldsymbol{q}}_i + 2\bar{\boldsymbol{q}}_i^T \boldsymbol{b} - \lambda\eta \|\bar{\boldsymbol{q}}_i\|_2^2).$$

where we denote $\bar{\boldsymbol{q}}_i$ as $\bar{\boldsymbol{q}}_i^{(k)}$ for conciseness, and $\hat{\boldsymbol{B}}$ as either the i.i.d. noise matrix $\boldsymbol{B}$ or symmetric noise matrix $\tilde{\boldsymbol{B}}$. Since $\|\bar{\boldsymbol{q}}_i\|_2 \leq N$, according to Lemma 1, we have the following inequalities:

$$2\bar{\boldsymbol{q}}_i^T \boldsymbol{b} \leq 2\|\bar{\boldsymbol{q}}_i\|_2\|\boldsymbol{b}\|_2 \leq 2N\|\boldsymbol{b}\|_2,$$

$$-\bar{\boldsymbol{q}}_i^T \hat{\boldsymbol{B}} \bar{\boldsymbol{q}}_i = -\sum_{(j,l)} \hat{B}_{jl}' \bar{q}_{ij} \bar{q}_{il}$$
$$\leq \left(\sum_{(j,l)} \hat{B}_{jl}'^2\right)^{\frac{1}{2}} \left(\sum_{(j,l)} \bar{q}_{ij}^2 \bar{q}_{il}^2\right)^{\frac{1}{2}}$$
$$= \|\hat{\boldsymbol{B}}\|_F \left[\left(\sum_j \bar{q}_{ij}^2\right)^2\right]^{\frac{1}{2}}$$
$$\leq \|\hat{\boldsymbol{B}}\|_F N^2.$$

Thus, we have

$$\mathbb{E}[L(\bar{\boldsymbol{q}}_i) - L(\boldsymbol{q}_i^*)] = \mathbb{E}[(\boldsymbol{q}_i^{*T}\hat{\boldsymbol{B}}\boldsymbol{q}_i - 2\boldsymbol{q}_i^{*T}\boldsymbol{b} + \lambda\eta\,\|\boldsymbol{q}_i^*\|_2^2) +$$
$$(-\bar{\boldsymbol{q}}_i^T\hat{\boldsymbol{B}}\bar{\boldsymbol{q}}_i + 2\bar{\boldsymbol{q}}_i^T\boldsymbol{b} - \lambda\eta\,\|\bar{\boldsymbol{q}}_i\|_2^2)]$$
$$= \mathbb{E}(-\bar{\boldsymbol{q}}_i^T\hat{\boldsymbol{B}}\bar{\boldsymbol{q}}_i + 2\bar{\boldsymbol{q}}_i^T\boldsymbol{b} - \lambda\eta\,\|\bar{\boldsymbol{q}}_i\|_2^2)$$
$$\leq \mathbb{E}\left[\|\hat{\boldsymbol{B}}\|_F N^2\right] + \mathbb{E}\left[2N\|\boldsymbol{b}\|_2\right]$$

By Lemma 2, $\bar{\boldsymbol{q}}_i^{(1)}, \bar{\boldsymbol{q}}_i^{(2)}, \bar{\boldsymbol{q}}_i^{(3)}$ satisfy

$$\mathbb{E}[L(\bar{\boldsymbol{q}}_i^{(1)}) - L(\boldsymbol{q}_i^*)] \leq \frac{(cd^2 + 2cd + 1)(N^2\sqrt{2}d + 2N\sqrt{2d})}{\varepsilon}$$
$$\leq \frac{\sqrt{2}cdN\left[(N + \frac{2\sqrt{d}}{d})(d^2 + 2d + \frac{1}{cd})\right]}{\varepsilon}$$
$$\leq \frac{\sqrt{2}cdN\left[(N + \frac{2\sqrt{d}}{d})(d+1)^2\right]}{\varepsilon}$$

$$\mathbb{E}[L(\bar{\boldsymbol{q}}_i^{(2)}) - L(\boldsymbol{q}_i^*)] \leq \frac{N^2\sqrt{2}d\Delta_2}{\varepsilon_2} + \frac{2N\sqrt{2d}\Delta_1}{\varepsilon_1}$$
$$\leq \frac{\sqrt{2}N^2cd^3(1-\alpha_0)}{\varepsilon_2} + \frac{4Ncd\sqrt{2d}}{\varepsilon_1}$$
$$= \frac{\sqrt{2}cdN\left[\varepsilon_1 Nd^2(1-\alpha_0) + 4\varepsilon_2\sqrt{d}\right]}{\varepsilon_1\varepsilon_2}.$$
$$= \frac{\sqrt{2}cdN\left[\frac{1}{\beta_2}Nd^2(1-\alpha_0) + \frac{4}{\beta_1}\sqrt{d}\right]}{\varepsilon}$$

$$\mathbb{E}[L(\bar{\boldsymbol{q}}_i^{(3)}) - L(\boldsymbol{q}_i^*)]$$
$$\leq \frac{N^2\sqrt{2}d\Delta_2}{\varepsilon_2} + \frac{2N\sqrt{2d}\Delta_1}{\varepsilon_1}$$
$$\leq \frac{\frac{\sqrt{2}}{2}N^2cd^2(1+d)(1-\alpha_0)}{\varepsilon_2} + \frac{4Ncd\sqrt{2d}}{\varepsilon_1}$$
$$= \frac{\sqrt{2}cdN\left[\frac{1}{2}\varepsilon_1 Nd(1+d)(1-\alpha_0) + 4\varepsilon_2\sqrt{d}\right]}{\varepsilon_1\varepsilon_2}.$$
$$= \frac{\sqrt{2}cdN\left[\frac{1}{2\beta_2}Nd(1+d)(1-\alpha_0) + \frac{4}{\beta_1}\sqrt{d}\right]}{\varepsilon}$$

Similarly, we can derive the bound for $\bar{\boldsymbol{q}}_i^{(4)}$ as

$$\mathbb{E}[L(\bar{\boldsymbol{q}}_i^{(4)}) - L(\boldsymbol{q}_i^*)] \leq \mathbb{E}\left[2N\|\boldsymbol{b}\|_2\right] \leq \frac{4\sqrt{2}cdN\sqrt{d}}{\varepsilon} \qquad \blacksquare$$

## J. Proof of Corollary 1

*Proof:* We first prove that $\gamma_1 > \gamma_2$ by presenting $(\gamma_1 - \gamma_2)$ is positive, which is defined as

$$\gamma_1 - \gamma_2 \propto N\left[d^2\left(\frac{\beta_2 + \alpha_0 - 1}{\beta_2}\right) + 2d + 1\right]$$
$$+ \frac{\beta_1(d+1)^2 - 1}{2d\beta_1} = f(\boldsymbol{\theta}).$$

We compute the partial derivative of $f(\boldsymbol{\theta})$ w.r.t $N$ as follows

$$\frac{\partial f(\boldsymbol{\theta})}{\partial N} = d^2\left(\frac{\beta_2 + \alpha_0 - 1}{\beta_2}\right) + 2d + 1,$$

where $\boldsymbol{\theta}$ denotes the parameters including $\alpha_0, d, N, \beta_1$ and $\beta_2$. By ensuring $\alpha_0 \geq 1 - \beta_2$, we have $\frac{\partial f(\boldsymbol{\theta})}{\partial N} \geq 0$, which ensures $f$ is monotonically increasing with the value of $N \in (0, +\infty)$. Let $d \geq \sqrt{1/\beta_1} - 1$, we have $\lim_{N\to 0^+} f(\boldsymbol{\theta}) \geq 0$. Thus, the function $f(\boldsymbol{\theta})$ is strictly positive in the domain of $N \in (0, +\infty)$, which in turn proves $\gamma_1 > \gamma_2$. Since $\frac{1}{2}d(1+d) \leq d^2$ for all $d \in \mathbb{N}^+$, we have $\gamma_2 \geq \gamma_3$, and it is clear that $\gamma_3 \geq \gamma_4$ for all $\beta_1 \in [0, 1]$.

For all $\alpha_0, \beta_1, \beta_2 \in [0, 1]$, $d \in \mathbb{N}^+$ and for all $N, c \in (0, +\infty)$

$$\gamma_1 - \gamma_4 \propto N(d+1) + \frac{2\sqrt{d}(d^2+1)}{d} > 0$$

Thus, we have $\gamma_1 > \gamma_4$, which completes the proof. $\blacksquare$

## K. Additional Experimental Results

### K.1. Parameter Settings

For the setting of hyper-parameters, the number of sub-datasets is set to $K = 10$. For all MF-based methods, the total and local iterations of model learning are set to 100 and 20, respectively. This means each party queries the local dataset 2000 times during the federated learning. The parameters of MF-based methods (i.e., including DPMF and our DPIMF solutions) are empirically set according to optimal results of hyperparameter searching (where we repeat each combination of possible hyperparameter values 10 times across all datasets) of different datasets involved. Specifically, the regularization parameters $\lambda$ is set to 0.07 for ML-10M, and 1.0 for YahooMusic and Amazon, as ML-10M is denser and thus requires less regularization to fit its richer interactions. The number of latent factors $d$ is set to 20, 16 and 8 for ML-10M, YahooMusic and Amazon dataset, respectively. For two neighbor-based methods (i.e., DPLCF and LDPICF), we set the number of neighbors to 100 according to the best results of parameter tuning.

### K.2. Privacy Budget Allocation

We split the budget as $\varepsilon_k = \beta_k \varepsilon$ with $\sum_{k=1}^3 \beta_k = 1$ (Section 4.2) and tune $(\beta_1, \beta_2, \beta_3)$ by grid search on DPIMF$_{\text{sym}}$, where all coefficients are perturbed. The allocation follows the sensitivities (Section 4.1): the quadratic term dominates, with $\Delta_2 = O(d^2)$ over its $d(d+1)/2$ entries, against $\Delta_1 = O(d)$ and $\Delta_3 = O(1)$. Since the per-term noise scales as $\Delta_k/(\beta_k\varepsilon)$, $\beta_2$ should be the largest—yet not too large, as starving the linear term also hurts. Table 3 bears this out: HR@10 peaks at $(0.1, 0.8, 0.1)$, while both under-weighting (e.g., $(0.8, 0.1, 0.1)$) and over-weighting ($(0.05, 0.9, 0.05)$) the quadratic term are worse. We adopt $(0.1, 0.8, 0.1)$ throughout; at $\alpha_0 = 1$ (DPIMF$_{\text{opt}}$), $\Delta_2 = \Delta_3 = 0$, so only the linear term is noised and the split is immaterial.

*Table 3.* HR@10 on ML-10M (DPIMF$_{sym}$, $\varepsilon$=5) under different budget splits $(\beta_1, \beta_2, \beta_3)$.

| $(\beta_1, \beta_2, \beta_3)$ | HR@10 |
|---|---|
| (0.1, 0.1, 0.8) | 0.28 |
| (0.8, 0.1, 0.1) | 0.29 |
| (1/3, 1/3, 1/3) | 0.36 |
| (0.2, 0.6, 0.2) | 0.39 |
| **(0.1, 0.8, 0.1)** | **0.40** |
| (0.05, 0.9, 0.05) | 0.38 |

### K.3. Evaluation Metrics

For recommendation effectiveness, we mainly consider two metrics, namely Hit Ratio (HR) and Normalized Discounted Cumulative Gain (NDCG). HR measures whether the test item appears in the target user's recommendation list, and NDCG measures the ranked position of the hit. HR is defined as follows:

$$\text{HR@k} = \frac{\sum_{u \in \boldsymbol{U}} |I_p(u)@k \cap I_a(u)|}{\sum_{u \in \boldsymbol{U}} |I_a(u)|}, \tag{23}$$

where $I_p(u)@k$ is the set of top k items in the ranked recommendation list. The symbol $I_a(u)$ represents the set of items the user $u$ has interacted with. NDCG is given by

$$\text{NDCG@k} = \frac{1}{|U|} \sum_{u \in U} \frac{\text{DCG}_u@k}{\text{IDCG}_u@k}, \tag{24}$$

$$\text{DCG}_u@k = \sum_{\text{idx}=1}^{k} \frac{2^{rel_{\text{idx}}} - 1}{\log_2(\text{idx} + 1)}, \tag{25}$$

where $rel_{\text{idx}} \in \{0, 1\}$ indicates whether there is an interaction between user $u$ and the idx-th item in the ranked list, and $\text{IDCG}_u@k$ is the ideal $\text{DCG}_u@k$ computed on the recommendation list sorted by $rel_{\text{idx}}$ in descending order. The larger values of HR and NDCG, the better recommendation quality.

### K.4. The Effect of Complementary Loss

In this section, we evaluate the effectiveness of the proposed complementary loss in Section 4 by comparing the recommendation performance of DPIMF$_{str}$, DPIMF$_{com}$, DPIMF$_{sym}$ and DPIMF$_{opt}$.

We first assess the effectiveness of complementary loss in limiting the magnitude of noises added to the loss function on all three datasets. We record the distributions of the noises in the second-order coefficients $\tilde{\boldsymbol{B}}$ (or $\boldsymbol{B}$) in DPIMF$_{str}$, DPIMF$_{com}$, DPIMF$_{sym}$, respectively. For DPIMF$_{com}$, DPIMF$_{sym}$, we set $\alpha_0 = 0.8$, $\varepsilon = 0.1$, $\beta_1 = 0.1$, $\beta_2 = 0.8$ and $\beta_3 = 0.1$. The distributions in ML-10M, YahooMusic and Amazon datasets are reported in Figures 8(a), (c) and (e). It is clearly shown in the figure that the noise distribution of DPIMF$_{sym}$ is sharper than other two schemes, while the curve of DPIMF$_{str}$ is the flattest. Such shapes convey that the variance of the noise in DPIMF$_{sym}$

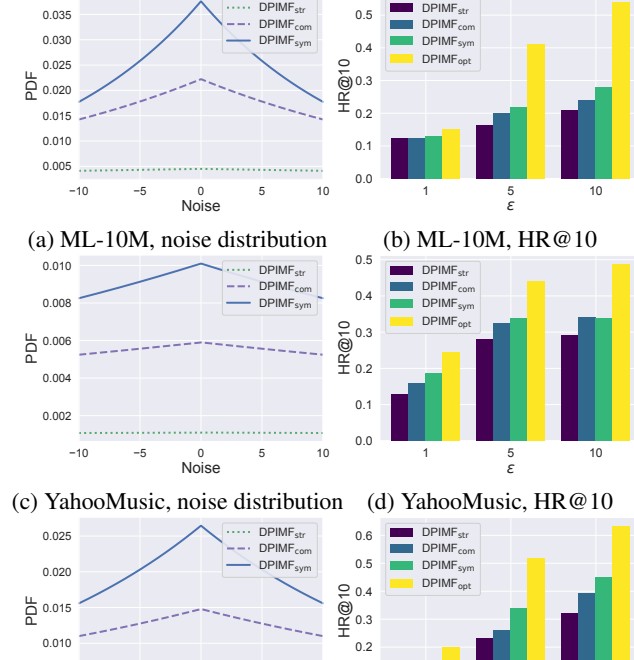

(a) ML-10M, noise distribution    (b) ML-10M, HR@10

(c) YahooMusic, noise distribution   (d) YahooMusic, HR@10

(e) Amazon, noise distribution    (f) Amazon, HR@10

*Figure 8.* Performance of DPIMF methods on noise distribution and HR@10.

is the lowest. And the variance of the noise in DPIMF$_{str}$ is higher than that in DPIMF$_{com}$. The differences of noise distributions indicate that the magnitude of the noise added to DPIMF$_{sym}$ is lower than the other two schemes, and in turn it enjoys an improvement in recommendation accuracy.

In order to validate the above claim, we evaluate the HRs of the recommendation lists generated by the three schemes on ML-10M, YahooMusic and Amazon datasets, respectively. With varying privacy budget $\varepsilon$ from 1 to 10, the test results are reported in Figure 8 (b), (d) and (f).

It is obvious that DPIMF$_{str}$ suffers the largest utility loss. When $\varepsilon > 1$, the other three schemes consistently outperform DPIMF$_{str}$ in HR across all datasets. The performance on the YahooMusic and Amazon datasets are shown in Figures 8 (d) and (f). DPIMF$_{com}$ outperforms DPIMF$_{sym}$ in some cases (e.g., the HR on Amazon when $\varepsilon = 5$). This is because the noise variances of the two methods are both relatively large on YahooMusic and Amazon, as shown in Figures 8(c) and (e).

Note that DPIMF$_{opt}$ always performs the best, achieving significantly higher HRs under different privacy budgets across all datasets. On Amazon dataset with $\varepsilon = 10$, the HR of DPIMF$_{opt}$ reaches 0.64, which is 32%, 25% and 19% higher than that of DPIMF$_{str}$, DPIMF$_{com}$ and DPIMF$_{sym}$, respectively. The results are consistent with the conclusions

of utility analysis. By setting $\alpha_0$ to 1, the effect of the change in data is completely offset. The ramification of this setting is that the second-order coefficients are free from perturbation. It not only reduces the injected noise, but also saves the privacy budget. More privacy budget is utilized to inject less noise into the objective function. As a consequence, the utility of recommendation increases dramatically, which explains the superiority of DPIMF$_{\text{opt}}$ shown in Figures 8 (b), (d) and (f).

### K.5. Comparing with other schemes

With varying privacy budgets from 1 to 10, the HRs and NDCGs of the competitors on ML-10M, YahooMusic and Amazon datasets are reported in Figure 9.

On ML-10M, as shown in Figures 9(a) and (b), the HR of DPIMF-IS is on average $18.3\%$ and $8.3\%$ higher than DPLCF and LDPICF. In NDCG, DPIMF-IS is $12.1\%$ and $5.4\%$ higher than DPLCF and LDPICF. The improvements become more significant on YahooMusic as shown in Figures 9(c) and (d). When $\varepsilon = 3$, the HR and NDCG of DPIMF-IS are $26\%, 20\%$ higher than DPLCF, and $35\%, 22\%$ higher than LDPICF, respectively. Furthermore, this accuracy gap even reaches $50\%$ around both in HR and NDCG on Amazon dataset.

The KNN-based methods suffer from the difficulty in finding proper neighbors in data with high sparsity level. This is demonstrated in Figures 9(e) and (f). The figures tell that the HRs and NDCGs of DPLCF and LDPICF are just around 0.1 on Amazon, the sparsest dataset of the three. The results demonstrate that the DPLCF and LDPICF are far from practical on extremely sparse datasets. On the contrary, the HR and NDCG of DPIMF on the same dataset increase steadily when increasing the privacy budget. When $\varepsilon = 10$, the HR and NDCG of DPIMF exceed 0.6 and 0.5, respectively. The utility loss is reduced to around 0.2. This result verifies the advantage of DPIMF in dealing with highly sparse datasets.

To verify the effectiveness of DPIMF in the scenario of learning private user profile matrix, we conducted the tests on subdatasets split by items using the same hyper-parameters settings. The results are shown in Figure 10. Compared to the case of learning item profiles, on the ML-10M dataset, all algorithms showed significant improvement, where DPIMF-IS approaches the ground truth more quickly when $\varepsilon > 5$. This improvement is attributed to the higher per-user interactions as shown in Table 1, allowing each party to exploit more information to counter the noise during local training. On the YahooMusic dataset, as the data per user is much sparser than the data per item, causing greater fluctuations in the algorithm's performance. This is due to the limited data amplifying the impact of noise, which is similarly observed in the results on Amazon dataset.

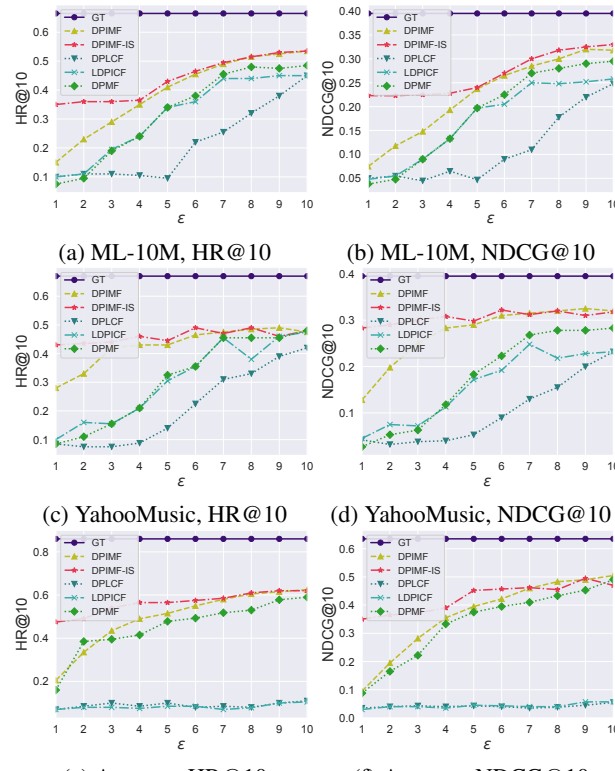

(a) ML-10M, HR@10    (b) ML-10M, NDCG@10

(c) YahooMusic, HR@10    (d) YahooMusic, NDCG@10

(e) Amazon, HR@10    (f) Amazon, NDCG@10

*Figure 9.* Overall results on HR@10 and NDCG@10 for solving the item profile matrix.

## L. Sensitivity Analysis

We provide the sensitivity analysis in Section 4. Let $c = \max_{u \in \boldsymbol{U}, j \in \{1,2,...,d\}} |p_{uj}|$. By the norm constraint $\|\boldsymbol{q}_i\|_2 \leq \sqrt{1/\lambda}$ (Lemma 1), applied symmetrically to the user profiles, $c \leq \max_u \|\boldsymbol{p}_u\|_2 \leq \sqrt{1/\lambda}$; hence $c$ is a public constant and the sensitivities below are data-independent. For $\Delta$, the first-order term of Eq. (11), same as in Eq. (8), its coefficients satisfy

$$\left\| 2 \sum_{j=1}^d \left( \sum_{u \in \boldsymbol{U}_i} \boldsymbol{p}_u - \sum_{u \in \boldsymbol{U}_i'} \boldsymbol{p}_u \right)_j \right\|_1 \leq 2cd.$$

Thus, we define the sensitivity of the first-order term as

$$\Delta_1 = 2cd.$$

For the second-order term of Eq. (11), the following bound is hold

$$\left\| \sum_{1 \leq j, l \leq d} \left( \sum_{u \in \boldsymbol{U}_i} \boldsymbol{p}_u \otimes \boldsymbol{p}_u - \sum_{u \in \boldsymbol{U}_i'} \boldsymbol{p}_u \otimes \boldsymbol{p}_u \right. \right.$$
$$\left. \left. + \alpha_0 \sum_{u \in \tilde{\boldsymbol{U}}_i} \boldsymbol{p}_u \otimes \boldsymbol{p}_u - \alpha_0 \sum_{u \in \tilde{\boldsymbol{U}}_i'} \boldsymbol{p}_u \otimes \boldsymbol{p}_u \right)_{jl} \right\|_1$$
$$\leq \sum_{1 \leq j, l \leq d} (1 - \alpha_0) |(\boldsymbol{p}_v \otimes \boldsymbol{p}_v)_{jl}|$$
$$\leq \max_{u \in \boldsymbol{U}} ((1 - \alpha_0) \|\boldsymbol{p}_u \otimes \boldsymbol{p}_u\|_{1,1}) \leq (1 - \alpha_0) c^2 d^2.$$

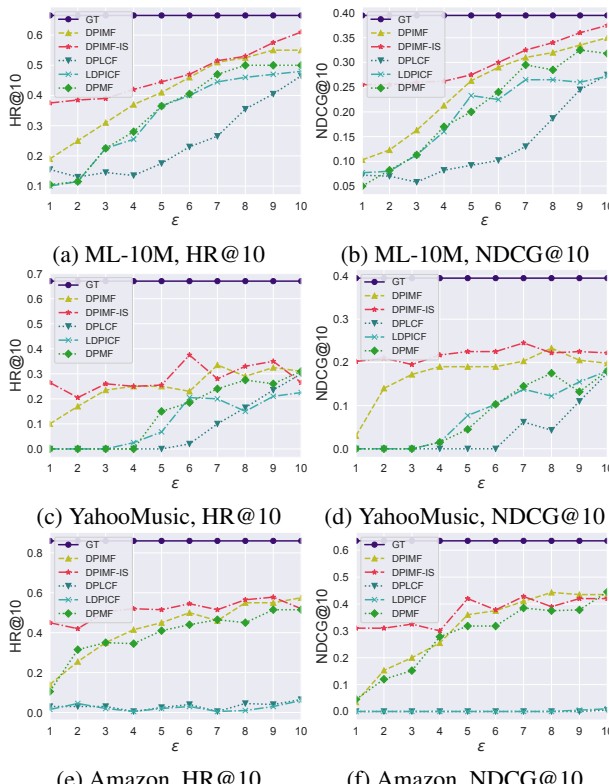

(a) ML-10M, HR@10    (b) ML-10M, NDCG@10

(c) YahooMusic, HR@10    (d) YahooMusic, NDCG@10

(e) Amazon, HR@10    (f) Amazon, NDCG@10

*Figure 10.* Overall results on HR@10 and NDCG@10 for solving the user profile matrix.

Therefore, the sensitivity of the second-order term is

$$\Delta_2 = (1 - \alpha_0)c^2 d^2.$$

For the regularization term $R(\boldsymbol{q}_i)$ of Eq. (10), we have

$$\left\| |\boldsymbol{U}_i| - |\boldsymbol{U}_i'| + \alpha_0|\tilde{\boldsymbol{U}}_i| - \alpha_0|\tilde{\boldsymbol{U}}_i'| \right\|_1 \leq 1 - \alpha_0$$

The sensitivity of $R(\boldsymbol{q}_i)$ is in turn defined as

$$\Delta_3 = 1 - \alpha_0.$$

Our second design is built upon the following observation. The coefficients of the polynomial in Eq. (11) are defined over complementary sets $\boldsymbol{U}_i$ and $\tilde{\boldsymbol{U}}_i$, implying the fact that they are independent of each other. Thus, the matrix $(\boldsymbol{p}_u \otimes \boldsymbol{p}_u)$ is symmetric.

From this observation, we can get a tighter bound of the sensitivity $\Delta_2$ as

$$\Delta_2 = \max_{u \in \boldsymbol{U}} \left( (1 - \alpha_0) \| \boldsymbol{p}_u \otimes \boldsymbol{p}_u \|_{\text{triu}} \right) \tag{26}$$

$$\leq \frac{c^2 d(d+1)(1 - \alpha_0)}{2}, \tag{27}$$

where $\| \boldsymbol{p}_u \otimes \boldsymbol{p}_u \|_{\text{triu}} = \sum_{j \leq l} |(\boldsymbol{p}_u \otimes \boldsymbol{p}_u)_{jl}|$ is computed on the values in the upper triangular elements of the matrix.

