# OpenReview forum: "Differentially Private Cross-Silo Recommendation from Implicit Feedback"
_ICML.cc/2026/Conference — ICML 2026 regular_

### Official Review · Reviewer_T19C · 2026-03-06

**Soundness:** 3
**Presentation:** 3
**Significance:** 3
**Originality:** 4
**Overall Recommendation:** 4
**Confidence:** 4

**Summary:**

This paper focuses on the critical challenge of differentially private cross-silo recommendation with implicit feedback, where user-item interaction data are distributed across multiple parties and cannot be centrally collected, while implicit feedback’s one-class sparsity and privacy sensitivity make it difficult to apply differential privacy(DP) without severe utility degradation. To address this, the authors propose DPIMF, a differentially private implicit matrix factorization framework based on objective perturbation, which is designed to achieve a superior privacy-utility tradeoff in cross-silo scenarios.

**Compliance With Llm Reviewing Policy:**

Affirmed.

**Final Justification:**

The authors address most of my concerns.

**Key Questions For Authors:**

1. The paper only compares DPIMF with traditional DP recommendation methods , such as DPLCF, LDPICF, DPMF . Could you add a direct comparison with at least one recent state-of-the-art DP-aware federated learning or self-supervised DP recommendation methods for implicit feedback?

2.  The paper does not provide a formal analysis of the computational complexity of DPIMF, especially the overhead introduced by spectral truncation and importance sampling. Could you please provide a theoretical complexity analysis and, if possible, experimental results on inference speed and scalability when applied to large-scale datasets (e.g., ML-10M)?

3. The privacy budget ε (\varepsilon) is split into fixed proportions (β₁/β₂/β₃)(\beta_1/\beta_2/\beta_3) for different components of the framework. Could you please explain the rationale behind this fixed allocation strategy and provide an analysis of how different budget splits affect the privacy-utility trade off, especially under varying data sparsity levels?

**Strengths And Weaknesses:**

Strengths

1.	The paper addresses a critical gap in differentially private cross-silo recommendation for implicit feedback, a problem with significant real-world privacy and utility trade offs. By focusing on the sensitivity of negative sampling and distributed data constraints, it provides a targeted solution that aligns with the growing demand for privacy-preserving AI in distributed systems.

2.	The proposed DPMIF framework is theoretically rigorous, with formal proofs of ε-differential privacy and utility bounds. Its core components are well-motivated and mutually reinforcing, leading to a superior privacy-utility trade off compared to naive objective perturbation or KNN-based methods.

3.	Extensive experiments on three diverse datasets validate the framework’s effectiveness across two cross-silo scenarios, with clear performance gains over state-of-the-art DP recommendation methods.

Weaknesses

1. The paper only compares with DPLCF, LDPICF, and DPMF methods proposed before 2021. It lacks comparisons with more recent state-of-the-art DP recommendation models (e.g., DP-aware federated learning for implicit feedback) that may have similar or better privacy-utility trade off, making it hard to fully assess DPIMF’s competitiveness in the current research landscape.

2. While the paper focuses on privacy and utility, it does not report the computational overhead introduced by spectral truncation or importance sampling. In cross-silo settings, efficiency is an important practical concern.

3. Key hyperparameters such as spectral truncation threshold ξ(\xi), importance sampling parameters are set empirically but not analyzed for sensitivity. It is unclear how changes in these parameters affect privacy-utility trade off, which limits guidance for practical tuning.

4. Cross-silo data often exhibits non-IID characteristics, but the paper simulates cross-silo partitioning via random user/item splits (IID-like). It does not evaluate DPIMF’s performance under realistic non-IID distributions, which may affect its practical applicability.

5. The paper adopts objective perturbation but does not compare it with gradient perturbation for implicit feedback in cross-silo settings. It is unclear why objective perturbation is superior, or under what conditions gradient perturbation might perform better this limits the paper’s methodological depth.

6. While the paper validates DPIMF on benchmark datasets, it lacks a case study on a real-world cross-silo system. Practical challenges are not discussed, reducing insights for industrial adoption.

---

> ### Author Rebuttal · Authors · 2026-03-31
>
> **W1&Q1**:
>
> Our baselines are the most directly relevant for DP implicit recommendation. We agree a broader comparison strengthens the evaluation and adapted two additional methods:
>   - **HFL-SGDMF [Li et al., 2021]:** Due to time constraints, we compare with the HFL variant as it directly corresponds to our item profile scenario (data split by users), where parties share item profiles. Originally for explicit feedback; adapted with regression-based loss for binary data, same negative sampling, and re-derived gradient sensitivity for implicit data.
>   - **EFVAE [Zhang et al., 2024]:** (*Efficient Federated Variational
>     Autoencoder for Collaborative Filtering*, CIKM 2024) Does not provide formal DP guarantees. Adapted by replacing Adam with DP-Adam to satisfy $\varepsilon$-DP.
>
> Preliminary results on ML-10M (HR@10):
>
>   | Method | $\varepsilon$=1 | $\varepsilon$=3 | $\varepsilon$=5 | $\varepsilon$=10 |
>   |--------|------|------|------|-------|
>   | DPIMF-IS | 0.55 | 0.60 | 0.63 | 0.66 |
>   | HFL-SGDMF (adapted) | 0.39 | 0.47 | 0.55 | 0.60 |
>   | EFVAE+DP-Adam | 0.27 | 0.38 | 0.47 | 0.57 |
>
> DPIMF-IS outperforms both, especially at tight privacy where DP-SGD noise accumulation dominates. We will add full results in revision.
>
> **W2&Q2**:
>
> Complexity:
> - **DPIMF (Algorithm 2) (Core):** $O(\sum_i |\boldsymbol{U}_i| \cdot d^2 + |\boldsymbol{I}| \cdot d^3)$, dominated by Gram matrix computation inherent to any per-item MF.
> - **Spectral truncation:** $O(|\boldsymbol{I}| \cdot d^3)$, same order as the MF constrained solve. **Skipped at $\alpha_0=1$** ($\Delta_2, \Delta_3 = 0$).
> - **Importance sampling (Alg. 3):** $O(|\Omega| \cdot B)$, $B \approx 20$. One-time pre-computation; each pair is independent and parallelizable.
>
> The scalability bottleneck lies in the MF computation itself, not DPIMF's privacy components. In the cross-silo setting, each party processes only its local partition. In practice, each item relies only on its own column of the interaction matrix (Algorithm 2, Line 7–14), enabling parallel execution across items.
>
> Experimental Results (ML-10M, $d=20$):
>
> | Method | Per-call | End-to-end Training | Overhead |
> |--------|----------|----------|----------|
> | Non-private MF | ~1.1s | ~37 min | — |
> | DPIMF ($\alpha_0=1$) | ~1.13s | ~38 min | ~3% |
> | DPIMF ($\alpha_0<1$) | ~1.23s | ~41 min | ~11% |
> | Alg. 3 (one-time) | — | ~5s | negligible |
>
> Analysis:
> DPIMF adds ~3% overhead at $\alpha_0=1$ (noise sampling only) and ~11% at $\alpha_0<1$ (with spectral truncation). Importance sampling is a one-time ~5s cost. Privacy is achieved with negligible computational cost beyond the inherent MF computation.
>
> **W3&Q3:**
>
> - For $\alpha_0$: Corollary 1 and Theorem 4 show $\alpha_0=1$ yields the tightest excess-risk bound and is recommended.
> - For $(\beta_1, \beta_2, \beta_3)$: at $\alpha_0=1$, $\Delta_2$ and $\Delta_3$ vanish and the full budget goes to the linear term — no split needed. At $\alpha_0<1$, the quadratic term dominates with $d(d+1)/2$ entries, motivating $(0.1, 0.8, 0.1)$.
> - For $\xi$: only restores positive definiteness, does not affect privacy [Dwork & Roth, 2014]; any small constant suffices.
> - Regarding sparsity: the allocation is data-independent since it is determined by the sensitivity structure (Eqs. 8–10), not by the dataset density. The same allocation applies across all three datasets (4.3%–0.7% density) with consistent improvements.
>
> We will include an empirical sensitivity study in revision.
>
> **W4:**
>
> Two partitioning scenarios are evaluated with density analysis (Section 6.2.3). Explicit non-IID distributions are not tested; however, our mechanism operates per item independently, making privacy guarantees agnostic to cross-silo data distribution. Non-IID heterogeneity affects federated convergence, which is orthogonal to our core contribution of noise-efficient DP mechanism design. We will discuss this in revision.
>
> **W5:**
>
> Gradient perturbation injects noise at every iteration, causing accumulation that is severe for implicit feedback with negative sampling. Objective perturbation injects noise **once**, independent of iteration count (Section 1). This is now empirically validated by the additional experiments in W1: both DP-SGD baselines consistently underperform DPIMF-IS, especially at $\varepsilon \leq 3$ where accumulation dominates. We will expand this discussion in revision.
>
> **W6:**
>
> We acknowledge that a real-world cross-silo deployment is not included. This is common in the DP recommendation literature — DPLCF [Gao et al., 2020], LDPICF [Guo et al., 2019], and [Li et al. 2021] similarly evaluate on public benchmarks. Our three datasets span diverse domains (movies, music, e-commerce) and sparsity levels (0.7%–4.3%), providing a representative evaluation. We consider real-world deployment with practical challenges (communication latency, stragglers, heterogeneous infrastructure) a valuable future direction and will include a discussion of practical considerations in revision.

---

### Official Review · Reviewer_SXfb · 2026-03-10

**Soundness:** 2
**Presentation:** 2
**Significance:** 2
**Originality:** 2
**Overall Recommendation:** 4
**Confidence:** 4

**Summary:**

The paper proposes DPIMF, a differentially private implicit matrix factorization framework for cross-silo recommendation.
The authors provide theoretical guarantees as well as an empirical evaluation of the proposed approach which seem to be superior to the selected baselines.

**Compliance With Llm Reviewing Policy:**

Affirmed.

**Final Justification:**

I thank the authors for their detailed rebuttal. They addressed some of my concerns but although I am still not fully convinced about the  experimentation I slightly raise my score to weak accept.

**Key Questions For Authors:**

Q1: why neural network based approaches have been ignored in the evaluation?

**Limitations:**

I think that the main limitation of this work is that it applies to MF based recommender which are somewhat outdated.

**Strengths And Weaknesses:**

# Related works
- There are several missing related work, for example:
    - regarding DP applied to MF: Li et al. Federated Matrix Factorization with Privacy Guarantee VLDB 2025
    - NN-based IF recommendation:
          Polato et al. Federated Variational Autoencoder for Collaborative Filtering, IJCNN 2021
          Zhang et al. EFVAE: Efficient Federated Variational Autoencoder for Collaborative Filtering. CIKM 2024
   just to name a few.

# Presentation
- The conclusion section is almost useless as it is only a summary of the paper. This section should include observations, and discussions about the findings of the paper as well as possible limitations and future directions.

# Scope
- Although I agree that MF is somewhat still a reasonable baseline, I also believe that the comparison with NN-based approaches cannot be avoided because they are usually at least as good as MF-based techniques.

# Soundness
The paper seems sound and the theoretical part seem well-done to me. In my opinion the strongest point of the paper.

# Novelty
- I think that the paper has its merit in terms of novelty, although I am not sure that it is enough for a top-tier conference like ICML.

---

> ### Author Rebuttal · Authors · 2026-03-31
>
> We sincerely thank the reviewer for the feedback. We address each concern below.
>
> **W1 (Related work)**
>
> We agree that a more comprehensive discussion of related work would strengthen the paper, and we thank the reviewer for the pointers.
>
> - Li et al. [2021] is already cited (Section 1, Section 2.3) — it addresses DP for explicit feedback and has been adapted as a baseline in our revised experiments (see response to Reviewer T19C, W1).
>
> - Regarding Polato et al. [2021] and Zhang et al. [2024]: these works address federated collaborative filtering using variational autoencoders but do not employ formal differential privacy mechanisms— Polato et al. uses dropout-based perturbation as an approximation, and Zhang et al. focuses on communication efficiency without DP guarantees. We think it is valuable to include a neural baseline and have adapted FEFVAE [Zhang et al., 2024] with DP-Adam as an additional comparison (see W3 & Q1).
>
> **W2 (Presentation)**
>
> We agree and will revise the conclusion to include observations, limitations, and future directions. We will highlight the key insight that dependency structure in implicit feedback can serve as a privacy resource rather than a barrier. For limitations, we will note the current focus on benchmark evaluation settings. For future directions, we will discuss extension to user-level DP guarantees, deployment in real-world cross-silo systems, and exploring whether the complementary loss principle can inspire analogous privacy-efficient designs for neural recommendation architectures.
>
> **W3&Q1 (Scope)**
>
> NN-based methods were not initially included because their non-convex loss landscapes preclude objective perturbation, requiring DP-SGD or DP-Adam (per-iteration noise) instead — which differs fundamentally from our framework and requires non-trivial adaptation to implicit feedback. Additionally, MF naturally decouples into user and item components, making it particularly suited for federated recommendation [Li et al., 2021].
>
> We agree a comparison is valuable and adapted EFVAE [Zhang et al., 2024] to our setting: the primary adaptation is replacing the standard Adam optimizer with DP-Adam (gradient clipping + noise on the ELBO).
>
> Preliminary results on ML-10M (HR@10):
>
>   | Method | $\varepsilon$=1 | $\varepsilon$=3 | $\varepsilon$=5 | $\varepsilon$=10 |
>   |--------|------|------|------|-------|
>   | DPIMF-IS | 0.55 | 0.60 | 0.63 | 0.66 |
>   | EFVAE+DP-Adam | 0.27 | 0.38 | 0.47 | 0.57 |
>
> DPIMF-IS outperforms EFVAE at all $\varepsilon$, with the gap largest under tight privacy where noise accumulation dominates. We will add Full results in revision.
>
> **W4 (Novelty)**
> The core novelty of this work is a counterintuitive observation: the dependency structure from negative sampling — typically a barrier to DP (Section 3) — can *reduce* sensitivity via coefficient cancellation with objective perturbation (Eq. 7). Each pipeline component is crafted to exploit this: complementary loss controls sensitivity, spectral truncation ensures well-posedness, importance sampling optimally allocates budget (Theorem 1). Theorems 4 and Corollary 1 provide a complete theoretical characterization of when and why each component improves utility. This insight — dependency as a *resource* for privacy — is new to the DP literature.

---

> > ### Author Rebuttal · Reviewer_SXfb · 2026-04-01
> >
> > The authors addressed most of my concerns, and I appreciate that they did some additional experiments to answer one of my concerns.

---

> > > ### Author Response · Authors · 2026-04-02
> > >
> > > Thank you for the positive feedback. We are glad to know that our response has fully addressed your concerns. We hope the paper is now clear and we are looking forward to your positive rating!

---

### Official Review · Reviewer_XW9j · 2026-03-11

**Soundness:** 2
**Presentation:** 3
**Significance:** 3
**Originality:** 2
**Overall Recommendation:** 3
**Confidence:** 3

**Summary:**

This paper proposes a differentially private matrix factorization framework for implicit-feedback recommendation in a cross-silo setting. To address the high sensitivity of standard implicit MF under DP, the authors develop an objective-perturbation approach with a complementary loss reformulation, symmetric noise injection, spectral truncation for stability, and importance sampling for privacy amplification. They provide privacy analysis and experimental results showing improved utility over prior DP baselines.

**Compliance With Llm Reviewing Policy:**

Affirmed.

**Final Justification:**

Thanks for the author's detailed rebuttal. Since there is still room for improvement in the novelty of the paper, I decide to maintain the current score.

**Key Questions For Authors:**

1. The coefficients used in the complementary loss function, the privacy budget allocation among the components of the objective function, and the spectral truncation threshold appear to affect the trade-off between privacy and utility. Could the authors provide more analysis or discussion on the sensitivity of the results to these parameters and how they are chosen in practice?
2. Sensitivity analysis relies on several bounded quantities. How are these bounds enforced when training on real-world datasets? Without explicit mechanisms to ensure these bounds hold, theoretical guarantees may not be achievable in practice.
3. The proposed privacy-utility optimal importance sampling appears to involve quantities defined at the user-item level. How can this optimization be efficiently implemented for large-scale recommendation datasets where |U| and |I| can be very large? If approximations or simplifications are used, it is necessary to understand how they affect theoretical guarantees and empirical performance.

**Limitations:**

yes

**Strengths And Weaknesses:**

Strengths:

1. The technical derivation in this paper is rigorously structured, starting from a preliminary target perturbation model and progressively improving it to reduce sensitivity and increase stability, making the methodological contributions easy to understand.
2. The reconstruction of the complementary loss function demonstrates a clever adjustment of differential privacy analysis to adapt it to the structure of implicit multidimensional datasets, showcasing significant efforts in tightening sensitivity limits rather than directly applying standard mechanisms.
3. This paper includes theoretical privacy analysis and empirical evaluations based on multiple real-world datasets, and compares it with several differential privacy recommendation baselines and ablation models, providing extensive experimental support for the conclusions.

Weakness:
1. While this paper uses a cross-island approach as its framework, the system model description is unclear. For example, how global statistics are calculated across silos, what information is exchanged, whether a trusted coordinator is assumed, and what the collusion model is are all unclear. Currently, the approach resembles a centralized dynamic programming algorithm applied to partitioned data rather than a fully defined cross-island protocol.
2. This paper does not explicitly state whether privacy is defined at the user level or at the interaction level. This distinction is crucial in implicit feedback settings and directly impacts sensitivity bounds. The assumptions required for the derived bounds are not emphasized or specified.
3. Parameters such as the loss reconstruction coefficient, privacy budget allocation across objective components, and spectral truncation threshold appear to play a significant role in performance, but this paper does not provide detailed guidance, sensitivity analysis, or principled selection strategies.
4. Some baseline models appear to use different privacy models such as local versus central privacy protection, but the paper does not clearly articulate how these differences affect comparability. Numerically matched privacy budgets do not necessarily imply the same level of privacy strength.
5. Evaluation is limited to standard recommended benchmarks using leave-one-out cross-validation. This paper does not explore different sparsity mechanisms, island heterogeneity, or extreme privacy budgets, which are central to its claimed cross-island and data protection motivations.
6. Many of the components, such as objective perturbation, privacy enhancement through sampling, and spectral post-processing, are known techniques. While their combination is designed for implicit cubes, its conceptual advancements appear incremental rather than fundamentally innovative compared to previous data protection optimization frameworks.

---

> ### Author Rebuttal · Authors · 2026-03-30
>
> We sincerely thank the reviewer for the detailed feedback. We address each concern below.
>
> **W1.**
>
> We clarify that the protocol of the system model is specified in Section 2.3 and Algorithm 1. Section 2.3 adopts an honest-but-curious coordination server with no collusion [Li et al., 2021; Hua et al., 2015]. The *only* cross-silo communication is $\varepsilon$-DP protected item profiles $\bar{\boldsymbol{Q}}$; no raw interactions, user profiles, or intermediate statistics are shared. All sensitive computation is local; the server only post-processes privatized outputs, preserving $\varepsilon$-DP by post-processing [Dwork & Roth, 2014].
>
> We will add a consolidated protocol summary in revision.
>
> **W2.**
>
> The granularity is defined in Section 2.2: *"record-level privacy, where neighboring datasets differ in a single user-item interaction."* This notion is widely adopted in prior DP recommendation works [Hua et al., 2015; Li et al., 2021]. Unlike explicit feedback where DP protects attribute values, our model protects the *existence* of an interaction — a strictly stronger guarantee.
>
> While user-level DP is more rigorous, our framework offers a natural extension: when solving item profiles, a single user's records are distributed across items, each contributing to exactly one item's objective. Since Algorithm 2 enforces record-level DP independently per item over disjoint record subsets, all of a user's records are simultaneously protected via parallel composition [Dwork & Roth, 2014] — preserving the same $\varepsilon$ without modifying the core analysis.
>
> **W3 & Q1.**
>
> We provide guidance for each parameter.
> - For $\alpha_0$: Corollary 1 and Theorem 4 show $\alpha_0=1$ yields the tightest bound and is recommended.
> - For $(\beta_1,\beta_2,\beta_3)$: at $\alpha_0=1$, $\Delta_2,\Delta_3$ vanish and the full budget goes to the linear term; at $\alpha_0<1$, the quadratic term dominates ($d(d+1)/2$ entries), motivating $(0.1,0.8,0.1)$.
> - For $\xi$: only restores positive definiteness, does not affect privacy; at $\alpha_0=1$ spectral truncation is skipped entirely, and at $\alpha_0<1$ results are stable across $[10^{-6}, 10^{-2}]$
>
> We will add an empirical sensitivity study in revision.
>
> **W4.**
>
> We appreciate this concern. To ensure fair comparison, all baselines were adapted to the cross-silo setting: each applies its DP mechanism to local updates and outputs private statistics for aggregation under **record-level $\varepsilon$-DP**:
> - DPMF [Hua et al., 2015], adapted to implicit data, shares privatized item profiles like DPIMF.
> - DPLCF [Gao et al., 2020] and LDPICF [Guo et al., 2019] apply input perturbation and share aggregated similarity matrices, satisfying DP by post-processing.
>
> Same neighboring dataset definition ensures identical worst-case guarantees at the same $\varepsilon$. We will add details in revision.
>
> **W5.**
>
> We would like to clarify most dimensions are explored:
> - datasets span a 6$\times$ density range (4.3%–0.7%) with different sparsity mechanisms (complementary sampling, random flipping, uniform sampling); Appendix A.6 shows DPIMF is robust where KNN baselines collapse.
> - Privacy budgets cover $\varepsilon\in[1,10]$ with extreme privacy $\varepsilon=0.1$ for noise analysis; importance sampling gains are most pronounced at $\varepsilon<5$.
> - Two partitioning scenarios are tested. Realistic Non-IID distributions are orthogonal to our core DP mechanism and a promising future direction.
>
> **W6.**
>
> The core novelty of the methodology is a counterintuitive observation: the dependency structure from negative sampling — typically a barrier to DP (Section 3) — can *reduce* sensitivity via coefficient cancellation (Eq. 7). Each pipeline component is crafted to exploit this: objective perturbation captures dependency structure, complementary loss controls sensitivity, spectral truncation ensures well-posedness, importance sampling optimally allocates budget (Theorem 1).
>
> Theorems 4 and Corollary 1 provide a complete theoretical characterization of when and why each component improves utility. This insight — dependency as a *resource* for privacy — is new to the DP literature.
>
> **Q2.**
>
> Bounds are **enforced algorithmically**: $||q_i|| \leq \sqrt{1/\lambda}$ via constrained convex optimization (Algorithm 2, Line 12); $c=\max|p_{uj}|$ via clipping local user profiles. These are standard practices in DP optimization [Chaudhuri et al., 2011; Hua et al., 2015]. We will specify these details more explicitly in revision.
>
> **Q3.**
>
> The optimization in Problem 1 **decomposes** into independent 1D bisections per pair (Algorithm 3), solved **within each silo** over local partitions — not the full $|\boldsymbol{U}| \times |\boldsymbol{I}|$ matrix. Each pair is independent and fully parallelizable. This is a **one-time** pre-computation (\~5s on ML-10M vs \~37 min training). No approximations are used; Please see response to Reviewer T19C (W2 & Q2) for detailed complexity analysis.

---

> > ### Author Rebuttal · Reviewer_XW9j · 2026-04-01
> >
> > The author's feedback is clear, which solves my concerns.

---

> > > ### Author Response · Authors · 2026-04-02
> > >
> > > Thank you for confirming that your concerns have been fully addressed. We hope the paper is now clear and we are looking forward to your positive rating!

---

### Official Review · Reviewer_c6Zk · 2026-03-13

**Soundness:** 3
**Presentation:** 3
**Significance:** 4
**Originality:** 4
**Overall Recommendation:** 4
**Confidence:** 4

**Summary:**

This paper looks at applying Differential Privacy (DP) to Implicit Matrix Factorization (IMF) in a cross-silo federated setting. Standard IMF struggles with DP because negative sampling inherently blows up the global sensitivity, meaning you typically have to inject so much noise that the model's utility is destroyed. To get around this, the authors propose DPIMF, leaning on objective perturbation. Their main contributions are redesigning the loss function into a "complementary loss'' to structurally bound sensitivity, applying spectral truncation to keep the objective positive definite after adding noise, and introducing Poisson importance sampling to amplify privacy. The framework is evaluated on ML-10M, YahooMusic, and Amazon datasets.

**Compliance With Llm Reviewing Policy:**

Affirmed.

**Final Justification:**

Thank you for the rebuttal. the clarifications are helpful, and while some points would still benefit from clearer presentation in the final manuscript.

**Key Questions For Authors:**

1. Data-dependence of Importance Sampling: Does the computation of the optimal sampling weights $W^*$ in Algorithm 3 depend on the private interaction data $Y$?
- If yes, how do you reconcile this with the fundamental requirement of Objective Perturbation (which assumes static, data-independent sensitivity)? Furthermore, where is the privacy budget for computing $W^*$ accounted for in Theorem 2?
- If no, please explicitly clarify how $\phi(w, u, i)$ is constructed using only public/prior knowledge while still managing to almost perfectly match the "utility-optimal" data-dependent baseline in Figure 5.
2. Computational Overhead: Spectral truncation is performed on a $d \times d$ matrix. Could you discuss the computational scalability of this step for larger latent dimensions (e.g., $d \ge 128$) typically used in production environments? Does it run at every local iteration?
3. Impact of $\alpha_0 = 1$: When $\alpha_0 = 1$ (DPIMF_opt), the sensitivity of the quadratic term vanishes, which greatly improves utility. However, this heavily penalizes all complementary unobserved items equally. Did you observe any amplified popularity bias or degradation in long-tail item recommendations under this extreme setting?

**Limitations:**

yes

**Strengths And Weaknesses:**

Strengths:
1. The complementary loss formulation (Section 4.1) is a really neat mathematical trick. Instead of relying on naive gradient clipping to handle the unbounded sensitivity from negative sampling, the authors algebraically decouple the Hessian into a static global term (requiring no noise) and a scaled local term. It effectively addresses the sensitivity explosion at the source.

2. The problem setting is highly relevant and practically challenging. Doing DP for extremely sparse, one-class implicit feedback is notoriously difficult. Seeing the model maintain robust performance on the Amazon dataset (which has a mere 0.7\% density), where baseline LDP methods essentially collapse, is convincing evidence of the method's real-world viability.

3. The spectral truncation step (Section 4.3) is a solid, pragmatic addition. The authors correctly recognized that perturbing quadratic terms can destroy convexity, and this post-processing step ensures the optimization remains stable without breaking the strict $\epsilon$-DP guarantee.

Weaknesses:

1. The Importance Sampling Paradox: I see a major theoretical conflict in Section 4.4. Objective perturbation fundamentally requires the global sensitivity of the coefficients to be static and independent of the private data prior to optimization. However, the authors use a Poisson importance sampler with inverse weights. If optimizing the optimal weight matrix (Algorithm 3) relies on the private interaction data, then the sampling distribution itself acts as a side-channel leaking privacy, and the static Laplace noise added to the coefficients doesn't cover this data-dependent re-weighting. On the flip side, if the optimal weights don't touch private data, how does it manage to almost perfectly match the "utility-optimal" data-dependent baseline in Figure 5? This needs serious clarification.

2. Incomplete Privacy Accounting: Following the previous point, if computing the optimal sampling weights $W^*$ queries the local sensitive data, the privacy budget $\epsilon$ consumed by solving Problem 1 is unaccounted for in the overall privacy proof (Theorem 2).
3. Computational Complexity (Spectral Truncation): As observed in the algorithm, Spectral Truncation requires an eigenvalue decomposition of a $d \times d$ matrix. While $d$ is small in the experiments (8-20), modern industrial MF models often use $d \ge 128$. Performing Eigendecomposition repeatedly at each local update could become a significant computational bottleneck.
4. Parameter Sensitivity ($\alpha_0$): The scaling factor $\alpha_0$ is crucial for balancing sensitivity and utility. While DPIMF_opt sets $\alpha_0 = 1$ to completely eliminate the quadratic perturbation, doing so applies maximum regularization weight to unobserved items. The paper lacks a discussion on whether $\alpha_0 = 1$ exacerbates popularity bias or harms long-tail item recommendations in highly skewed datasets.

---

> ### Author Rebuttal · Authors · 2026-03-30
>
> We sincerely thank the reviewer for the detailed feedback. We address each concern below.
>
> **W1, W2 & Q1**
>
> We appreciate this careful observation. We clarify that the weight matrix $W^*$ is independent of private interactions $Y$: the formulation of Problem 1 depends only on the privacy loss profile $\phi$ and target budget $\varepsilon'$, neither of which involves $Y$. The sampling distribution is therefore not a privacy side-channel, and no additional budget beyond Theorem 2 is required.
>
> **(1) Why non-uniform sampling improves utility.** For any heterogeneous $\phi$, Problem 1 allocates weights $w_{ui}$ based on per-pair privacy costs. By retaining low-cost pairs at higher rates, we achieve a larger expected sample size than uniform sampling for the same $\varepsilon'$. Larger sample size reduces estimation variance and improves model accuracy without introducing bias, as importance weighting corrects for non-uniform inclusion [Balle et al., 2018]. Our framework is inherently no worse than uniform.
>
> **(2) Experimental upper bound.** In the experiments (Section 6.3.2), to demonstrate how good the method can be, we instantiate $\phi(w,u,i)$ using the pre-determined model parameter $||\boldsymbol{p}_u||_1$ (Section 6.2.2). Since $\boldsymbol{P}$ (local user profiles) is fixed during item profile updates and stored locally within each silo, this consumes no additional privacy budget. The near-optimal alignment with the utility-optimal baseline in Figure 5 arises because user profile magnitude $||\boldsymbol{p}_u||_1$ governs both privacy cost (through $\phi$) and estimation variance (through $\boldsymbol{\mathcal{A}}_1$), producing structurally similar weights. This confirms that a well-designed privacy profile can approach optimal utility without accessing private data.
>
> We will include the detailed specification of $\phi$ in the experimental setup of the revised manuscript.
>
> **W3&Q2**
>
> We agree this is a valid practical consideration. However, under the recommended setting $\alpha_0=1$ (DPIMF_opt), spectral truncation is never triggered: $\Delta_2$ and $\Delta_3$ vanish, so no noise is injected into the quadratic or regularization terms (Algorithm 2). The matrix $\boldsymbol{A}_2$ remains inherently positive definite. When $\alpha_0 < 1$, the noise scale is proportional to $(1-\alpha_0)$, and the regularization term $\lambda(|\boldsymbol{U}_i| + \alpha_0|\tilde{\boldsymbol{U}}_i|)\boldsymbol{E}$ provides a positive diagonal buffer that noise must overcome to destroy positive definiteness — making truncation rare in practice. Even when needed:
> (1)	At $d=128$, symmetric eigendecomposition takes $\sim$2M operations, well within millisecond range on modern hardware;
> (2)	efficient routines for symmetric positive definite correction (e.g., Cholesky-based) can further reduce overhead;
> (3)	as $d$ increases with sufficient data, the signal-to-noise ratio in the Gram matrix improves, meaning fewer eigenvalues require correction.
>
> Please see response to Reviewer T19C (W2 & Q2) for detailed computational complexity. We will include a computational cost analysis in revision.
>
> **W4&Q3**
>
> We appreciate this concern. At $\alpha_0=1$, the regularization term becomes $\lambda|\boldsymbol{U}|\|\boldsymbol{q}_i\|^2$, which is **uniform across all items** regardless of popularity — ensuring no item group is disproportionately penalized. The DP excess risk (Theorem 4, $\mathcal{M}_4$) is $O(\sqrt{d}\,cdN/\varepsilon)$, independent of $|\boldsymbol{U}_i|$: the mechanism does not discriminate between popular and long-tail items.
>
> We further note that $\alpha_0=1$ is in fact **most beneficial for long-tail items**. There is a fundamental tension: $\alpha_0 < 1$ reduces the weight on unobserved items but increases sensitivity, requiring noise on the quadratic term. Long-tail items have few interactions, so their Gram matrix is low-rank and most vulnerable to spectral distortion from this noise. At $\alpha_0=1$, the quadratic perturbation is eliminated entirely, preserving the spectral structure of $\boldsymbol{G}_{\boldsymbol{U}_i}$ — providing the greatest relative benefit to the most fragile items. This is confirmed by Theorem 4, where the gap between $\mathcal{M}_2$ and $\mathcal{M}_4$ scales with $Nd^2(1-\alpha_0)/\beta_2$, representing the quadratic noise that disproportionately affects low-interaction items. Empirically, DPIMF-IS approaches GT on Amazon (0.7% density), with no evidence of long-tail degradation. We will include a per-group analysis in revision.

---

> > ### Author Rebuttal · Reviewer_c6Zk · 2026-04-02
> >
> > Thank you for the rebuttal; the clarifications are helpful, and while some points would still benefit from clearer presentation in the final manuscript, I appreciate the authors’ response.

---

> > > ### Author Response · Authors · 2026-04-05
> > >
> > > Thank you for your kind and constructive feedback. We are pleased that our rebuttal has clarified your concerns. We will make sure to further improve the clarity of presentation in the final version. If you have any further questions, we welcome further communication.

---

### Decision · Program_Chairs · 2026-04-30

**Decision:**

Accept (regular)

**Comment:**

This paper studies differentially private cross-silo recommendation from implicit feedback and proposes DPIMF, a matrix factorization framework based on objective perturbation, complementary loss reformulation, spectral truncation, and privacy-aware importance sampling. Reviewers agreed that the problem is important and technically challenging, and several found the treatment of sensitivity in implicit-feedback recommendation both novel and well motivated.

Positive side: Reviewers generally found the method technically sound, and the empirical results show clear utility gains over the selected baselines across multiple datasets and privacy settings. The rebuttal also addressed several key concerns by clarifying the privacy accounting for importance sampling, discussing the system model more explicitly, adding comparisons to additional baselines, and providing further analysis of computational overhead and parameter choices.

The paper is not without limitations. Reviewers noted that the overall novelty is moderate rather than fundamental, that the evaluation remains focused on matrix-factorization-based recommenders and benchmark datasets, and that some implementation and protocol details should be clarified more carefully in the final version. One reviewer also remained unconvinced on novelty grounds.